# PREDICTING INDUCTIVE BIASES
# OF PRE-TRAINED MODELS

**Charles Lovering[†], Rohan Jha[†], Tal Linzen[§], Ellie Pavlick[†]**

[†] Brown University
Department of Computer Science
{charles_lovering@, rohan_jha@alumni., ellie_pavlick@} brown.edu

[§] New York University
Department of Linguistics and Center for Data Science
linzen@nyu.edu

## ABSTRACT

Most current NLP systems are based on a pre-train-then-fine-tune paradigm, in which a large neural network is first trained in a self-supervised way designed to encourage the network to extract broadly-useful linguistic features, and then fine-tuned for a specific task of interest. Recent work attempts to understand why this recipe works and explain when it fails. Currently, such analyses have produced two sets of apparently-contradictory results. Work that analyzes the representations that result from pre-training (via "probing classifiers") finds evidence that rich features of linguistic structure can be decoded with high accuracy, but work that analyzes model behavior after fine-tuning (via "challenge sets") indicates that decisions are often not based on such structure but rather on spurious heuristics specific to the training set. In this work, we test the hypothesis that the extent to which a feature influences a model's decisions can be predicted using a combination of two factors: The feature's *extractability* after pre-training (measured using information-theoretic probing techniques), and the *evidence* available during fine-tuning (defined as the feature's co-occurrence rate with the label). In experiments with both synthetic and naturalistic data, we find strong evidence (statistically significant correlations) supporting this hypothesis.

## 1 INTRODUCTION

Large pre-trained language models (LMs) (Devlin et al., 2019; Raffel et al., 2020; Brown et al., 2020) have demonstrated impressive empirical success on a range of benchmark NLP tasks. However, analyses have shown that such models are easily fooled when tested on distributions that differ from those they were trained on, suggesting they are often "right for the wrong reasons" (McCoy et al., 2019). Recent research which attempts to understand why such models behave in this way has primarily made use of two analysis techniques: *probing classifiers* (Adi et al., 2017; Hupkes et al., 2018), which measure whether or not a given feature is encoded by a representation, and *challenge sets* (Cooper et al., 1996; Linzen et al., 2016; Rudinger et al., 2018), which measure whether model behavior in practice is consistent with use of a given feature. The results obtained via these two techniques currently suggest different conclusions about how well pre-trained representations encode language. Work based on probing classifiers has consistently found evidence that models contain rich information about syntactic structure (Hewitt & Manning, 2019; Bau et al., 2019; Tenney et al., 2019a), while work using challenge sets has frequently revealed that models built on top of these representations do not behave as though they have access to such rich features, rather they fail in trivial ways (Dasgupta et al., 2018; Glockner et al., 2018; Naik et al., 2018).

In this work, we attempt to link these two contrasting views of feature representations. We assume the standard recipe in NLP, in which linguistic representations are first derived from large-scale self-supervised *pre-training* intended to encode broadly-useful linguistic features, and then are adapted for a task of interest via transfer learning, or *fine-tuning*, on a task-specific dataset. We test the

hypothesis that the extent to which a fine-tuned model uses a given feature can be explained as a function of two metrics: The *extractability* of the feature after pre-training (as measured by probing classifiers) and the *evidence* available during fine-tuning (defined as the rate of co-occurrence with the label). We first show results on a synthetic task, and second using state-of-the-art pre-trained LMs on language data. Our results suggest that probing classifiers can be viewed as a measure of the pre-trained representation's inductive biases: The more extractable a feature is after pre-training, the less statistical evidence is required in order for the model to adopt the feature during fine-tuning.

**Contribution.**   This work establishes a relationship between two widely-used techniques for analyzing LMs. Currently, the question of how models' internal representations (measured by probing classifiers) influence model behavior (measured by challenge sets) remains open (Belinkov & Glass, 2019; Belinkov et al., 2020). Understanding the connection between these two measurement techniques can enable more principled evaluation of and control over neural NLP models.

## 2  SETUP AND TERMINOLOGY

### 2.1  FORMULATION

Our motivation comes from McCoy et al. (2019), which demonstrated that, when fine-tuned on a natural language inference task (Williams et al., 2018, MNLI), a model based on a state-of-the-art pre-trained LM (Devlin et al., 2019, BERT) categorically fails on test examples which defy the expectation of a "lexical overlap heuristic". For example, the model assumes that the sentence *"the lawyer followed the judge"* entails *"the judge followed the lawyer"* purely because all the words in the latter appear in the former. While this heuristic is statistically favorable given the model's training data, it is not infallible. Specifically, McCoy et al. (2019) report that 90% of the training examples containing lexical overlap had the label "entailment", but the remaining 10% did not. Moreover, the results of recent studies based on probing classifiers suggest that more robust features are extractable with high reliability from BERT representations. For example, given the example *"the lawyer followed the judge"/"the judge followed the lawyer"*, if the model can represent that *"lawyer"* is the agent of *"follow"* in the first sentence, but is the patient in the second, then the model should conclude that the sentences have different meanings. Such semantic role information can be recovered at $> 90\%$ accuracy from BERT embeddings (Tenney et al., 2019b). Thus, the question is: Why would a model prefer a weak feature over a stronger one, if both features are extractable from the model's representations and justified by the model's training data?

Abstracting over details, we distill the basic NLP task setting described above into the following, to be formalized in the Section 2.2. We assume a binary sequence classification task where a *target* feature $t$ perfectly predicts the label (e.g., the label is 1 *iff* $t$ holds). Here, $t$ represents features which actually determine the label by definition, e.g., whether one sentence semantically entails another. Additionally, there exists a *spurious* feature $s$ that frequently co-occurs with $t$ in training but is not guaranteed to generalize outside of the training set. Here, $s$ (often called a "heuristic" or "bias" elsewhere in the literature) corresponds to features like lexical overlap, which are predictive of the label in some datasets but are not guaranteed to generalize.

**Assumptions.**   In this work, we assume there is a single $t$ and a single $s$; in practice there may be many $s$ features. Still, our definition of a feature accommodates multiple spurious or target features. In fact, some of our spurious features already encompass multiple features: the lexical feature, for example, is a combination of several individual-word features because it holds if one of a set of words is in the sentence. This type of spurious feature is common in real datasets: E.g., the hypothesis-only baseline in NLI is a disjunction of lexical features (with semantically unrelated words like "no", "sleeping", etc.) (Poliak et al., 2018b; Gururangan et al., 2018).

We assume that $s$ and $t$ frequently co-occur, but that only $s$ occurs in isolation. This assumption reflects realistic NLP task settings since datasets always contain some heuristics, e.g., lexical cues, cultural biases, or artifacts from crowdsourcing (Gururangan et al., 2018). Thus, our experiments focus on manipulating the occurrence of $s$ alone, but not $t$ alone: This means giving the model evidence against relying on $s$. This is in line with prior applied work that attempts to influence model behavior by increasing the evidence against $s$ during training (Elkahky et al., 2018; Zmigrod et al., 2019; Min et al., 2020).

## 2.2 Definitions

Let $\mathcal{X}$ be the set of all sentences and $S$ be the space of all sentence-label pairs $(x, y) \in \mathcal{X} \times \{0, 1\}$. We use $\mathcal{D} \subset S$ to denote a particular training sample drawn from $S$. We define two types of binary features: *target* ($t$) and *spurious* ($s$). Each is a function from sentences $x \in \mathcal{X}$ to a binary label $\{0, 1\}$ that indicates whether the feature holds.

**Target and spurious features.** The *target* feature $t$ is such that there exists some function $f : \{0, 1\} \rightarrow \{0, 1\}$ such that $\forall (x, y) \in S$, $f(t(x)) = y$. In other words, the label can always be perfectly predicted given the value of $t$.[1] A feature $s$ is *spurious* if it is not a *target* feature.

**Partitions of $S$.** To facilitate analysis, we partition $S$ in four regions (Figure 1). We define $S_{s\text{-only}}$ to be the set of examples in which the spurious feature occurs alone (without the target). Similarly, $S_{t\text{-only}}$ is the set of examples in which the target occurs without the spurious feature. $S_{\text{both}}$ and $S_{\text{neither}}$ are analogous. For clarity, we sometimes drop the $S_*$ notation (e.g., $s$-only in place of $S_{s\text{-only}}$).

$$S_{both} = \{(x, y) \,|\, t(x) = 1 \wedge s(x) = 1\}$$
$$S_{neither} = \{(x, y) \,|\, t(x) = 0 \wedge s(x) = 0\}$$
$$S_{t\text{-}only} = \{(x, y) \,|\, t(x) = 1 \wedge s(x) = 0\}$$
$$S_{s\text{-}only} = \{(x, y) \,|\, t(x) = 0 \wedge s(x) = 1\}$$

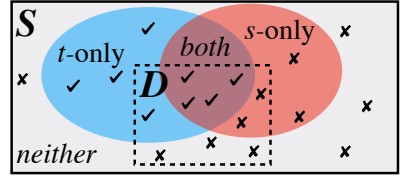

Figure 1: We partition datasets into four sections, defined by the features (spurious and/or target) that hold. We sample training datasets $D$, which provide varying amounts of *evidence* against the spurious feature, in the form of $s$-only examples. In the illustration above, the $s$-only rate is $\frac{2}{10} = 0.2$, i.e., 20% of examples in $D$ provide evidence that $s$ alone should not be used to predict $y$.

**Evidence from Spurious-Only Examples.** We are interested in spurious features which are highly correlated with the target during training. Given a training sample $\mathcal{D}$ and features $s$ and $t$, we define the **$s$-only example rate** as the evidence against the use of $s$ as a predictor of $y$. Concretely, $s$-only rate $= |D_{s\text{-only}}|/|D|$, the proportion of training examples in which $s$ occurs without $t$ (and $y = 0$).

**Use of Spurious Feature.** If a model has falsely learned that the spurious feature $s$ alone is predictive of the label, it will have a high error rate when classifying examples for which $s$ holds but $t$ does not. We define the **$s$-only error** to be the classifier's error on examples from $S_{s\text{-only}}$. When relevant, **$t$-only error**, **both error**, and **neither error** are defined analogously. In this work, "feature use" is a model's predictions consistency with that feature; we are not making a causal argument.

**Extractability of a Feature.** We want to compare features in terms of how *extractable* they are given a representation. For example, given a sentence embedding, it may be possible to predict multiple features with high accuracy, e.g., whether the word *"dog"* occurs, and also whether the word *"dog"* occurs as the subject of the verb *"run"*. However, detecting the former will no doubt be an easier task than detecting the latter. We use the prequential **minimum description length (MDL)** Rissanen (1978)–first used by Voita & Titov (2020) for probing–to quantify this intuitive difference.[2] MDL is an information-theoretic metric that measures how accurately a feature can be decoded and the amount of effort required to decode it. Formally, MDL measures the number of bits required to communicate the labels given the representations. Conceptually, MDL can be understood as a measure of the area under the loss curve: If a feature is highly *extractable*, a model trained to detect that feature will converge quickly to high accuracy, resulting in a low MDL. Computing MDL requires repeatedly training a model over a dataset labeled by the feature in question. To compute MDL$(s)$, we train a classifier (without freezing any parameters) to differentiate $S_{s\text{-only}}$ vs. $S_{\text{neither}}$, and similarly compute MDL$(t)$. See Voita & Titov (2020) for additional details on MDL.[3]

---

[1]Without loss of generality, we define $t$ in our datasets s.t. $t(x) = y, \forall x, y \in S$. We do this to iron out the case where $t$ outputs the opposite value of $y$.

[2]We observe similar overall trends when using an alternative metric based on validation loss (Appendix A.3).

[3]Note that our reported MDL is higher in some cases than that given by the uniform code (the number of sentences that are being encoded). The MDL is computed as a sum of the costs of transmitting successively

## 2.3 HYPOTHESIS

Stated using the above-defined terminology, our hypothesis is that a model's *use of the target feature* is modulated by two factors: The relative *extractability* of the target feature $t$ (compared to the spurious feature $s$), and the *evidence* from $s$-only examples provided by the training data. In particular, we expect that higher extractability of $t$ (relative to $s$), measured by $\mathrm{MDL}(s)/\mathrm{MDL}(t)$, will yield models that achieve better performance despite less training evidence.

## 3 EXPERIMENTS WITH SYNTHETIC DATA

Since it is often difficult to fully decouple the target feature from competing spurious features in practice, we first use synthetic data in order to test our hypothesis in a clean setting. We use a simple classifier with an embedding layer, a 1-layer LSTM, and an MLP with 1 hidden layer with tanh activation. We use a synthetic sentence classification task with $k$-length sequences of numbers as input and binary labels as output. We use a symbolic vocabulary $V$ with the integers $0 \ldots |V| - 1$. We fix $k = 10$ and $|V| = 50\mathrm{K}$. We begin with an initial training set of 200K, evenly split between examples from $S_{\mathrm{both}}$ and $S_{\mathrm{neither}}$. Then, varied across runs, we manipulate the evidence against the spurious feature (i.e., the $s$-only rate) by replacing a percentage $p$ of the initial data with examples from $S_{s\text{-only}}$ for $p \in \{0\%, 0.1\%, 1\%, 5\%, 10\%, 20\%, 50\%\}$. Test and validation sets consist of 1,000 examples each from $S_{\mathrm{both}}, S_{\mathrm{neither}}, S_{t\text{-only}}, S_{s\text{-only}}$. In all experiments, we set the spurious feature $s$ to be the presence of the symbol 2. We consider several different target features $t$ (Table 1), intended to vary in their extractability. Table 1 contains MDL metrics for each feature (computed on training sets of $200K$, averaged over 3 random seeds). We see some gradation of feature extractability, but having more features with wider variation would help solidify our results.[4]

| Target Feature | Description | MDL($s$) | MDL($t$) | Rel. MDL | Example |
|---|---|---|---|---|---|
| contains-1 | 1 occurs in sequence | 0.36 | 0.29 | 1.259 | 2 4 11 1 4 |
| prefix-dupl | Sequence begins with duplicate | 0.42 | 175.74 | 0.002 | 5 5 11 12 2 |
| adj-dupl | Adjacent duplicate in seq. | 0.37 | 242.20 | 0.001 | 11 12 3 3 2 |
| first-last | First number equals last number | 0.37 | 397.64 | 0.001 | 7 2 11 12 7 |

Table 1: Instantiations of the target feature $t$ in our synthetic experiments. The spurious feature $s$ is always the presence of the symbol 2. Features are intended to differ in how hard they are for an LSTM to detect given sequential input (measured by MDL per §2.2, reported in $k$-bits).

Figure 2 shows model performance as a function of $s$-only rate for each of the four features described above. Here, performance is reported using error rate (lower is better) on each partition ($S_{s\text{-only}}$, $S_{t\text{-only}}$, $S_{\mathrm{both}}$, $S_{\mathrm{neither}}$) separately. We are primarily interested in whether the relative extractability of the target feature (compared to the spurious feature) predicts model performance. We indeed see a fairly clear relationship between the relative extractability ($\mathrm{MDL}(s)/\mathrm{MDL}(t)$) and model performance, at every level of training evidence ($s$-only rate). For example, when $t$ is no less extractable than $s$ (i.e., contains-1), the model achieves zero error at an $s$-only rate of 0.001, meaning it learns that $t$ alone predicts the label despite having only a handful of examples that support this inference. In contrast, when $t$ is harder to extract than $s$ (e.g., first-last), the model fails to make this inference, even when a large portion of training examples provide evidence supporting it.

## 4 EXPERIMENTS WITH NATURALISTIC DATA

We investigate whether the same trend holds for language models fine-tuned with naturalistic data, e.g., grammar-generated English sentences. To do this, we fine-tune models for the linguistic acceptability task, a simple sequence classification task as defined in Warstadt & Bowman (2019),

---

longer blocks, using classifiers that are trained on previously transmitted data. The high MDL's are a result of overfitting by classifiers that are trained on limited data–and therefore, the classifiers have worse compression performance than the uniform baseline.

[4]Note, all models are ultimately able to learn to detect $t$ (achieve high test accuracy) on the both partition, but not on the $t$-only partition.

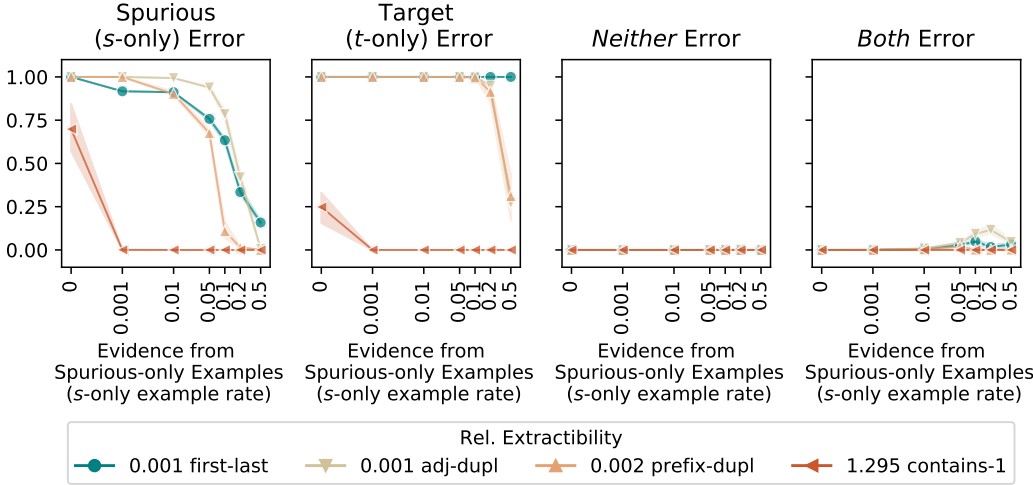

Figure 2: **Results on Synthetic Data**. Error on each partition of the test set, as a function of $s$-only rate. A model that has learned to use the target feature alone to predict the label will achieve zero error across all partitions. $s$-only and $t$-only error reach 0 quickly when $t$ is as easy to extract as $s$ (i.e., the relative extractability is 1). However, when $t$ is harder to extract than $s$ (rel. extractability $< 1$), performance lags until evidence from $s$-only examples is quite strong.

in which the goal is to differentiate grammatical sentences from ungrammatical ones. We focus on acceptability judgments since formal linguistic theory guides how we define the target features, and recent work in computational linguistics shows that neural language models can be sensitive to spurious features in this task (Marvin & Linzen, 2018; Warstadt et al., 2020a).

## 4.1 DATA

We design a series of simple natural language grammars that generate a variety of feature pairs $(s, t)$, which we expect will exhibit different levels of relative extractability $(\mathrm{MDL}(s)/\mathrm{MDL}(t))$. We focus on three syntactic phenomena (described below). In each case, we consider the target feature $t$ to be whether a given instance of the phenomenon obeys the expected syntactic rules. We then introduce several spurious features $s$ which we deliberately correlate with the positive label during fine-tuning. The **Subject-Verb Agreement (SVA)** construction requires detecting whether the verb agrees in number with its subject, e.g., *"the girls are playing"* is acceptable while *"the girls is playing"* is not. In general, recognizing agreement requires some representation of hierarchical syntax, since subjects may be separated from their verbs by arbitrarily long clauses. We introduce four spurious features: 1) lexical, grammatical sentences begin with specific lexical items (e.g., *"often"*); 2) length, grammatical sentences are longer; 3) recent-noun, verbs in grammatical sentences agree with the immediately preceding noun (in addition to their subject); and 4) plural, verbs in grammatical sentences are preceded by singular nouns as opposed to plural ones.

The **Negative Polarity Items (NPI)** construction requires detecting whether a negative polarity item (e.g., *"any"*, *"ever"*) is grammatical in a given context, e.g., *"no girl ever played"* is acceptable while *"a girl ever played"* is not. In general, NPIs are only licensed in contexts that fall within the scope of a downward entailing operator (such as negation). We again consider four types of spurious features: 1) lexical, in which grammatical sentences always include one of a set of lexical items (*"no"* and *"not"*); 2) length (as above); 3) plural, in which each noun in a grammatical sentence is singular, as opposed to plural; and 4) tense, in which grammatical sentences are in present tense.

Some verbs (e.g. *"recognize"*) require a direct object. However, in the right syntactic contexts (i.e., when in the correct syntactic relation with a $wh$-word), the object position can be empty, creating what is known as a "gap". E.g., *"I know what you recognized __"* is acceptable while *"I know that you recognized __"* is not. The **Filler-Gap Dependencies (GAP)** construction requires detecting

whether a sentence containing a gap is grammatical. For our GAP tasks, we again consider four spurious features (lexical, length, plural, and tense), defined similarly to above.

| Target | Spurious | Example |
|---|---|---|
| Subject agrees with verb | N before V is singular | [*both*] The piano teachers of the lawyer wound the handyman.
[*s*-only] *The piano teachers of the lawyer wounds the handyman. |
| NPI in down. -entailing context | Contains negation word | [*both*] No student who was wrong ever resigned.
[*s*-only] *The student who was not wrong ever resigned. |
| Correct filler-gap dependency | Main verb is in past tense | [*both*] I knew what he recognized ___ yesterday.
[*s*-only] *I knew what he recognized someone yesterday. |

Table 2: Examples of features used to generate fine-tuning sets with target/spurious features of varying extractability scores. Top examples show a case in which $t$ and $s$ both occur and the sentence is acceptable, and bottom examples show a case in which $s$ occurs without $t$ and the sentence is unacceptable. Only $s$ is highlighted since $t$ is often defined over the structure of the sentence (see text) and thus difficult to localize to a few tokens. Table 9 in the Appendix has *neither* examples.

The templates above (and slight variants) result in 20 distinct fine-tuning datasets, over which we perform our analyses (see Appendix for details). Table 2 shows several examples. For the purposes of this paper, we are interested only in the relative extractability of $t$ vs. $s$ given the pre-trained representation; we don't intend to make general claims about the linguistic phenomena *per se*. Thus, we do not focus on the details of the features themselves, but rather consider each template as generating one data point, i.e., an $(s, t)$ pair representing a particular level of relative extractability.

## 4.2 SETUP

We evaluate T5, BERT, RoBERTa, GPT-2 and an LSTM with GloVe embeddings (Raffel et al., 2020; Devlin et al., 2019; Liu et al., 2019b; Radford et al., 2019; Pennington et al., 2014).[5] Both T5 and BERT learn to perform well over the whole test set, whereas the GloVe model struggles with many of the tasks. We expect that this is because contextualized pre-training encodes certain syntactic features which let the models better leverage small training sets (Warstadt & Bowman, 2020). Again, we begin with an initial training set of 2000 examples, evenly split between *both* and *neither*, and then introduce $s$-only examples at rates of 0%, 0.1%, 1%, 5%, 10%, 20%, and 50%, using three random seeds each. Test and validation sets consist of 1000 examples each from $S_{\text{both}}, S_{\text{neither}}, S_{s\text{-only}}$. In the natural language setting, it is often difficult to generate $t$-only examples, and thus we cannot compute extractability of the target feature $t$ by training a classifier to distinguish $S_{t\text{-only}}$ from a random subset of $S_{\text{neither}}$, as we did in Section 3. Therefore, we estimate MDL by training a classifier to distinguish between examples from $S_{s\text{-only}}$ and examples from $S_{\text{both}}$. Using the simulated data from Section 3, we confirm that both methods ($S_{s\text{-only}}$ vs. $S_{\text{both}}$ and $S_{t\text{-only}}$ vs. $S_{\text{neither}}$) produce similar estimates of MDL($t$) (see Appendix). Per model, we filter out feature pairs for which the model could not achieve at least 90% accuracy on each probing task in isolation.[6]

## 4.3 RESULTS

For each $(s, t)$ feature pair, we plot the use of the spurious feature ($s$-only error) as a function of the evidence against the spurious feature seen in training ($s$-only example rate).[7] We expect to see the same trend we observed in our synthetic data, i.e., the more extractable the target feature $t$ is relative to the spurious feature $s$, the less evidence the model will require before preferring $t$ over $s$. To quantify this trend, we compute correlations between 1) the relative extractability of $t$ compared to $s$ and 2) the test F-score averaged across all rates and partitions of the data[8], capturing how readily the model uses (i.e., makes predictions consistent with the use of) the target feature.

---

[5]In pilot studies, we found that standard BOW and CNN-based models were unable solve the tasks.

[6]This control does not impact results: Appendix A.1.

[7]See Appendix for *both* error and *neither* error; both are stable and low in general.

[8]Initially, we used a more complicated metric based on the $s$ example rate required for the model to solve the test set. Both report similar trends and correlations. For posterity, we include details in the Appendix A.2.

|  | Absolute | | Relative ($t$ to $s$) | |
|---|---|---|---|---|
|  | Target | Spurious | Ratio | Difference |
| BERT | -0.72* | 0.65* | 0.79* | 0.96* |
| RoBERTa | -0.26 | 0.8* | 0.83* | 0.81* |
| T5 | -0.91* | 0.04 | 0.57* | 0.73* |
| GPT2 | -0.56* | 0.55* | 0.73* | 0.78* |
| GloVe | 0.12 | 0.44 | 0.14 | 0.10 |

(a) Spearman's $\rho$: MDL vs. Average Test F-score

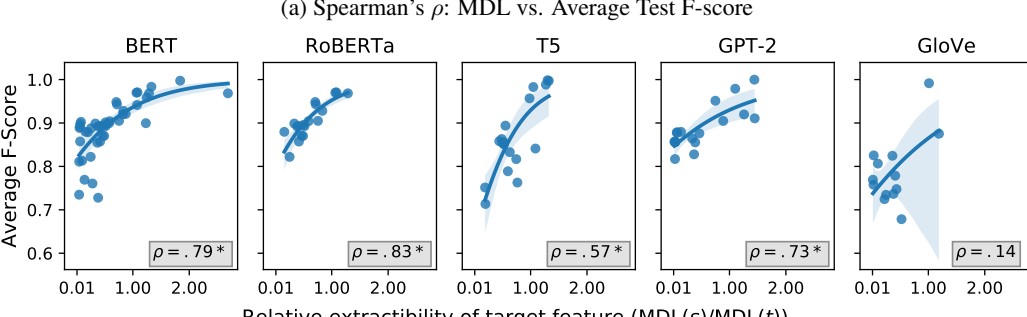

(b) Logistic regression plots of Average F-score vs. the relative extractability of $s, t$ via $(\mathrm{MDL}(s)/\mathrm{MDL}(t))$.

Figure 3: **Relative Extractability Correlates with Target Feature Use**. In (a) we show the Spearman's $\rho$ between the test F-Score vs measures of extractability of the $(s, t)$ pairs; * indicates significance. Relative extractability, whether ratio $(\mathrm{MDL}(s)/\mathrm{MDL}(t))$ or difference $(\mathrm{MDL}(s) - \mathrm{MDL}(t))$, explains learning behavior better than absolute extractability of either feature.

Figure 3 shows these correlations and associated scatter plots. We can see that relative extractability is strongly correlated with average test F-score (Figure 3a), showing high correlations for both BERT ($\rho = 0.79$) and T5 ($\rho = 0.57$). That is, the more extractable $t$ is relative to $s$, the less evidence the model requires before preferring $t$, performing better across all partitions. This relationship holds regardless of whether relative extractability is computed using a ratio of MDL scores or an absolute difference. We also see that, in most cases, the relative extractability explains the model's behavior better than does the extractability of $s$ or $t$ alone. For GloVe there is little variation in model behavior: For most of the 11/20 pairs on which the model is able to learn the task, it requires an s-only example rate of 0.5. Thus, the correlations are weak, but qualitative results appear steady (Figure 8 in Appendix A), following the pattern that when $s$ is easier to extract than $t$, more evidence is required to stop using $s$.

Figure 4 shows the performance curves for BERT and T5 (with others the in Appendix A), i.e., use of the spurious feature (s-only error) as a function of the evidence from s-only examples seen in training (s-only example rate). Each line corresponds to a different $s, t$ feature pair, and each data point is the test performance on a dataset with a given s-only example rate (which varies along the x-axis.) For pairs with high MDL ratios (i.e., when $t$ is actually easier to extract than $s$), the model learns to solve the task "the right way" even when the training data provides no incentive to do so: That is, in such cases, the models' decisions do not appear to depend on the spurious feature $s$ even when $s$ and the target feature $t$ perfectly co-occur in the fine-tuning data.

Figure 4 shows that T5 (compared to BERT) requires more data to perform well. This may be because we fine-tuned T5 with a linear classification head, rather than the text-only output on which it was pre-trained. We made this decision 1) because we had trouble training T5 in the original manner, and 2) using a linear classification head was consistent with the other model architectures.

## 5 DISCUSSION

**Our experimental results provide support for our hypothesis: The relative *extractability* of features given an input representation (as measured by information-theoretic probing tech-**

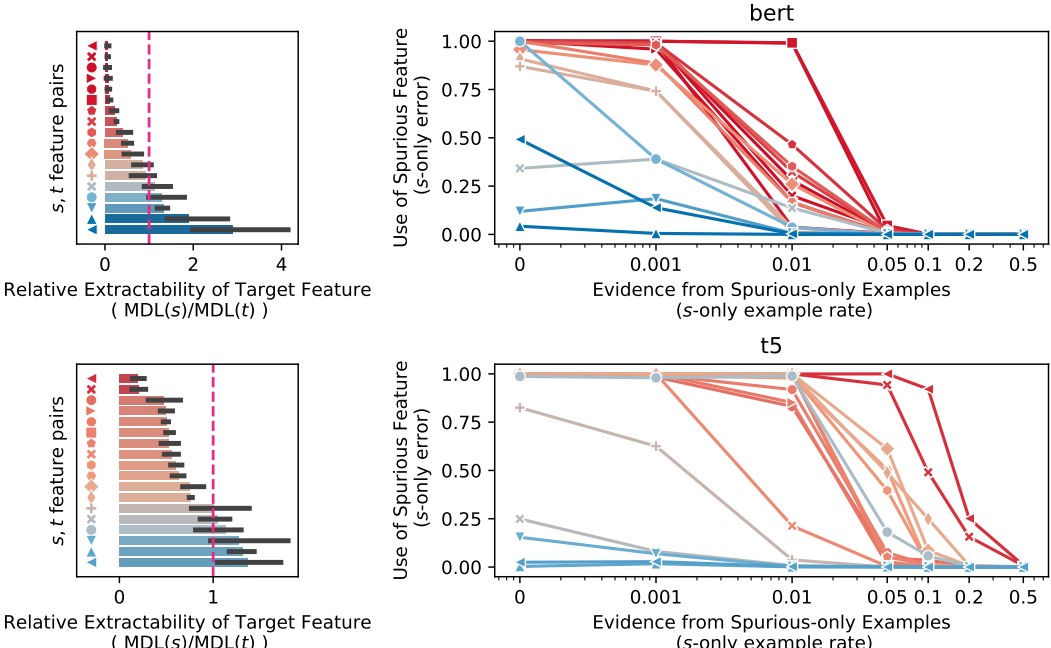

Figure 4: **Learning Curves for BERT & T5.** Curves show use of spurious feature ($s$-only accuracy) as a function of training evidence ($s$-only rate). Each line represents one $(s, t)$ pair (described in §4.1). Pairs vary in the relative extractability of $t$ vs. $s$ (measured by the ratio $\mathrm{MDL}(s)/\mathrm{MDL}(t)$ and summarized in the bar chart). When $t$ is much harder to extract relative to $s$ (lower ratios), the classifier requires much more statistical evidence during training (higher $s$-only rate) in order to achieve low error. We find similar patterns GPT2 and RoBERTa; see Appendix A for all the results.

**niques) is predictive of the decisions a trained model will make in practice.** In particular, we see evidence that models will tend to use imperfect features that are more readily extractable over perfectly predictive features that are harder to extract. This insight is highly related to prior work which has shown, e.g., that neural networks learn "easy" examples before they learn "hard" examples (Mangalam & Prabhu, 2019). Our findings additionally connect to new probing techniques which have received significant attention in NLP but have yet to be connected to explanations of or predictions about state-of-the-art models' decisions in practice.

**Fine-tuning may not uncover new features.** The models are capable of learning both the $s$ and $t$ features in isolation, so our experiments show that if the relative extractibility is highly skewed, one feature may hide the other – a fine-tuned model may not use the harder-to-extract feature. This suggests a pattern that seems intuitive but is in fact non-trivial: If one classifier does not pick up on a feature *readily enough*, another classifier (or, rather, the same classifier trained with different data) may not be sensitive to that feature *at all*. This has ramifications for how we view fine-tuning, which is generally considered to be beneficial because it allows models to learn new, task-relevant features. Our findings suggest that if the needed feature is not already extractable-enough after pretraining, fine-tuning may not have the desired effect.

**Probing classifiers can be viewed as measures of a pre-trained representation's inductive biases.** Analysis with probing classifiers has primarily focused on whether important linguistic features can be decoded from representations at better-than-baseline rates, but there has been little insight about what it would mean for a representations' encoding of a feature to be "sufficient". Based on these experiments, we argue that a feature is "sufficiently" encoded if it is as available to the model as are surface features of the text. For example, if a fine-tuned model can access features about a word's semantic role as easily as it can access features about that word's lexical identity, the model may need little (or no) explicit training signal to prefer a decision rule based on the former structural feature. The desire for models with such behavior motivates the development of architectures with explicit inductive biases (e.g., TreeRNNs). Evidence that similar generalization behavior

can result from pre-trained representations has exciting implications for those interested in sample efficiency and cognitively-plausible language learning (Warstadt & Bowman, 2020; Linzen, 2020). We note that this work has not established that the relationship between extractability and feature use is causal. This could be explored using intermediate task training (Pruksachatkun et al., 2020) in order to influence the extractability of features prior to fine-tuning for the target task; e.g., Merchant et al. (2020) suggests fine-tuning on parsing might improve the extractability of syntactic features.

## 6   RELATED WORK

Significant prior work analyzes the representations and behavior of pre-trained LMs. Work using probing classifiers (Veldhoen et al., 2016; Adi et al., 2017; Conneau et al., 2018; Hupkes et al., 2018) suggests that such models capture a wide range of relevant linguistic phenomena (Hewitt & Manning, 2019; Bau et al., 2019; Dalvi et al., 2019; Tenney et al., 2019a;b). Similar techniques include attention maps/visualizations (Voita et al., 2019; Serrano & Smith, 2019), and relational similarity analyses (Chrupała & Alishahi, 2019). A parallel line of work uses challenge sets to understand model behavior in practice. Some works construct evaluation sets to analyze weaknesses in the decision procedures of neural NLP models (Jia & Liang, 2017b; Glockner et al., 2018; Dasgupta et al., 2018; Gururangan et al., 2018; Poliak et al., 2018b; Elkahky et al., 2018; Ettinger et al., 2016; Linzen et al., 2016; Isabelle et al., 2017; Naik et al., 2018; Jia & Liang, 2017a; Linzen et al., 2016; Goldberg, 2019, and others). Others use such datasets to improve models' handling of linguistic features (Min et al., 2020; Poliak et al., 2018a; Liu et al., 2019a), or to mitigate biases (Zmigrod et al., 2019; Zhao et al., 2018; 2019; Hall Maudslay et al., 2019; Lu et al., 2020). Nie et al. (2020) and Kaushik et al. (2020) explore augmenting training sets with human-in-the-loop methods.

Our work is related to work on generalization of neural NLP models. Geiger et al. (2019) discusses ways in which evaluation tasks should be sensitive to models' inductive biases and Warstadt & Bowman (2020) discusses the ability of language model pre-training to encode such inductive biases. Work on data augmentation (Elkahky et al., 2018; Min et al., 2020; Zmigrod et al., 2019) is relevant, as the approach relies on the assumption that altering the training data distribution (analogous to what we call $s$-only rate in our work) will improve model behavior in practice. Kodner & Gupta (2020); Jha et al. (2020) discuss concerns about ways in which such approaches can be counterproductive, by introducing new artifacts. Work on adversarial robustness (Ribeiro et al., 2018; Iyyer et al., 2018; Hsieh et al., 2019; Jia et al., 2019; Alzantot et al., 2018; Hsieh et al., 2019; Ilyas et al., 2019; Madry et al., 2017; Athalye et al., 2018) is also relevant, as it relates to the influence of dataset artifacts on models' decisions. A still larger body of work studies feature representation and generalization in neural networks outside of NLP. Mangalam & Prabhu (2019) show that neural networks learn "easy" examples (as defined by shallow machine learning model performance) before they learn "hard" examples. Zhang et al. (2016) and Arpit et al. (2017) show that neural networks which are capable of memorizing noise nonetheless acheive good generalization performance, suggesting that such models might have an inherent preference to learn more general features. Finally, ongoing theoretical work characterizes the ability of over-parameterized networks to generalize in terms of complexity (Neyshabur et al., 2019) and implicit regularization (Blanc et al., 2020).

Concurrent work (Warstadt et al., 2020b) also investigates the inductive biases of large pre-trained models (RoBERTa), in particular, they ask when (at what amount of pre-training data) such models shift from a surface feature (what we call spurious features) to a linguistic feature (what we call a target feature). In our work, we focus on how to predict which of these two biases characterize the model (via relative MDL).

## 7   CONCLUSION

This work bears on an open question in NLP, namely, the question of how models' internal representations (as measured by probing classifiers) influence model behavior (as measured by challenge sets). We find that the feature extractability can be viewed as an inductive bias: the more extractable a feature is after pre-training, the less statistical evidence is required in order for the model to adopt the feature during fine-tuning. Understanding the connection between these two measurement techniques can enable more principled evaluation of and control over neural NLP models.

ACKNOWLEDGEMENTS

We would like to thank Michael Littman for helpful suggestions on how to better present our findings and Ian Tenney for insightful comments on a previous draft of this work. We also want to thank our reviewers were their detailed and helpful comments. This work is supported by DARPA under grant number HR00111990064. This research was conducted using computational resources and services at the Center for Computation and Visualization, Brown University.

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

# PREDICTING INDUCTIVE BIASES
# OF PRE-TRAINED MODELS

## A ADDITIONAL RESULTS

Figure 6, 7, 8, 9, 10 show additional results for all models over all partitions (*both* accuracy, *neither* accuracy, and F-score). These charts appear at the end of the Appendix.

Details on the MDL statistics are available in Table 3.

| | MDL($t$) | | | | |
|---|---|---|---|---|---|
| | BERT | GPT2 | GloVe | RoBERTa | T5 |
| gap_base_length | 443 | 614 | 2921 | 21 | 537 |
| gap_base_lexical | 223 | 337 | 1815 | 17 | 323 |
| gap_base_plural | 299 | 346 | 1664 | 14 | 401 |
| gap_base_tense | 278 | 417 | 1962 | 16 | 373 |
| gap_hard_length | 421 | 479 | 2845 | 18 | 493 |
| gap_hard_lexical | 292 | 341 | 1846 | 17 | 321 |
| gap_hard_none | 116 | 77 | 946 | 8 | 182 |
| gap_hard_plural | 322 | 440 | 1862 | 15 | 387 |
| gap_hard_tense | 354 | 353 | 1685 | 15 | 332 |
| npi_length | 300 | 240 | 952 | 16 | 273 |
| npi_lexical | 225 | 345 | 1249 | 12 | 245 |
| npi_plural | 292 | 335 | 1267 | 15 | 282 |
| npi_tense | 220 | 258 | 999 | 13 | 286 |
| sva_base_agreement | 113 | 463 | 1581 | 12 | 149 |
| sva_base_lexical | 150 | 629 | 5872 | 15 | 154 |
| sva_base_plural | 93 | 428 | 5122 | 14 | 166 |
| sva_hard_agreement | 173 | 579 | 1582 | 14 | 176 |
| sva_hard_length | 183 | 708 | 1466 | 17 | 197 |
| sva_hard_lexical | 181 | 684 | 1467 | 16 | 189 |
| sva_hard_plural | 192 | 698 | 1449 | 18 | 201 |
| | MDL($s$) | | | | |
| gap_base_length | 28 | 23 | 92 | 3 | 103 |
| gap_base_lexical | 297 | 425 | 452 | 12 | 241 |
| gap_base_plural | 3421 | 3794 | 1906 | 45 | 1924 |
| gap_base_tense | 159 | 141 | 710 | 9 | 235 |
| gap_hard_length | 42 | 16 | 46 | 4 | 98 |
| gap_hard_lexical | 313 | 493 | 410 | 12 | 246 |
| gap_hard_none | 144 | 85 | 957 | 9 | 179 |
| gap_hard_plural | 3200 | 3161 | 3987 | 37 | 1795 |
| gap_hard_tense | 131 | 131 | 694 | 7 | 198 |
| npi_length | 78 | 214 | 499 | 8 | 297 |
| npi_lexical | 46 | 48 | 475 | 6 | 123 |
| npi_plural | 15 | 22 | 1165 | 6 | 125 |
| npi_tense | 14 | 18 | 435 | 5 | 150 |
| sva_base_agreement | 305 | 348 | 1879 | 16 | 198 |
| sva_base_lexical | 12 | 14 | 162 | 5 | 85 |
| sva_base_plural | 77 | 199 | 535 | 11 | 174 |
| sva_hard_agreement | 319 | 835 | 1314 | 15 | 230 |
| sva_hard_length | 94 | 28 | 465 | 8 | 103 |
| sva_hard_lexical | 12 | 27 | 891 | 6 | 92 |
| sva_hard_plural | 168 | 274 | 695 | 14 | 256 |

Table 3: Summary of extractability (MDL in bits) for $t$ and $s$ for each template and each model.

### A.1 BEYOND ACCURACY?

For the transformer models, for 18/20 feature pairs, the models are able to solve all the spurious and target features in isolation during probing. (They do solve the test set in all cases–its that two

|        | Absolute | | Relative ($t$ to $s$) | |
|--------|----------|----------|-------|------------|
|        | Target   | Spurious | Ratio | Difference |
| BERT    | -0.51* | 0.72* | 0.83* | 0.95* |
| RoBERTa | -0.36  | 0.84* | 0.87* | 0.85* |
| T5      | -0.57* | 0.26  | 0.67* | 0.78* |
| GPT2    | -0.5*  | 0.66* | 0.8*  | 0.83* |
| GloVe   | 0.24   | 0.43  | 0.3   | 0.27  |

Table 4: Spearman's $\rho$: MDL vs. Average Test F-score: Correlations when Probing Accuracy is Not Controlled.

of the spurious features ended up being very difficult for the models.) During the reviews, we did not control for the cases where the model did not solve the probing task. These 2 extraneous points accentuate the lineplot curves, but do not change the character of the results (nor much adjust the correlations). In the paper, now, we control for accuracy by filtering out these cases. With or without this control, the accuracy provides no predictive power about the inductive biases. We present the correlations without filtering for these cases for consistency with the reviews (Table 4 above); we believe it is important to control for these cases because they could have acted as giveaways, where even accuracy might have worked.

## A.2 ALTERNATE METRIC: $s$-RATE$\star$

We initially used a different metric when computing the correlations to compact the lineplots. Rather than using the average test performance, we looked at the evidence required for the model to solve the test set. Both of these metrics conceptually capture what we are interested in, but the new one (simply averaging test performance) is much easier to understand, and captures the performance across all partitions. Here we report the correlations with this evidence required metric instead, which we called $s$-rate$\star$. Specifically, we defined it to be: $s$-rate$\star$ is the lowest $s$-only example rate at which the fine-tuned model achieves essentially perfect performance (F-score $> 0.99$) (see Figure 5a). Intuitively, $s$-rate$\star$ is the (observed) minimum amount of evidence from which the model infer that $t$ alone is predictive of the label. See Table 5b for the results.

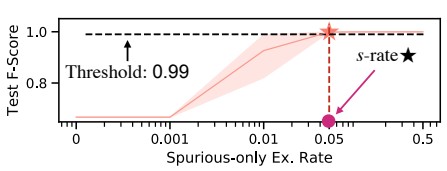

(a) Evidence Required: **$s$-rate$\star$**

|        | Absolute | | Relative ($t$ to $s$) | |
|--------|----------|----------|-------|------|
|        | Target   | Spurious | Ratio | Diff |
| BERT    | 0.68*  | -0.63* | -0.79* | -0.81* |
| RoBERTa | 0.04   | -0.69* | -0.71* | -0.69* |
| T5      | 0.81*  | -0.03  | -0.55* | -0.65* |
| GPT2    | -0.11  | -0.24  | -0.29  | -0.32  |
| GloVe   | 0.29   | -0.38  | -0.48  | -0.48  |

(b) Using Evidence Required (**$s$-rate$\star$**) instead of Average F-score. The correlations are negative instead of positive: As the extractibility increases, less evidence is required for the model to perform well.

## A.3 RESULTS USING AUC INSTEAD OF MDL

See Table 5. A metric similar to MDL for capturing the same intuition is the ***area under the validation loss curve (AUC)***. This metric is highly related to online MDL in computation.

## B IMPLEMENTATION DETAILS & REPRODUCIBLITY

Our code is available at:
`https://github.com/cjlovering/predicting-inductive-biases`.

| | Absolute | | Relative ($t$ to $s$) | |
| --- | --- | --- | --- | --- |
| | Target | Spurious | Ratio | Diff. |
| BERT | -0.61* | 0.61* | 0.71* | 0.67* |
| RoBERTa | -0.12 | 0.67* | 0.75* | 0.64* |
| T5 | -0.88* | 0.37 | 0.84* | 0.95* |
| GPT2 | -0.34 | 0.52* | 0.64* | 0.67* |
| GloVe | -0.5 | -0.63* | -0.23 | -0.14 |

Table 5: AUC. Each column presents the Spearman correlation between the given measure and the $s$-only rate at which the model first acquired the target feature. * indicates a significant correlation. A feature is considered acquired when its test-performance is above 0.99 F-score. All $(s, t)$ pairs (20 of 20 each) were eventually learned by T5, BERT, GPT2, and RoBERTa; 11 of 20 $(s, t)$ pairs were learned by the GloVe-based LSTM.

There are two major parts to this project in terms of reproducibility: (1) the data and (2) the model implementations. We describe the templates for the data below in Appendix D – the full details are in the project source. For the transformer models, we use Hugging Face for the implementations and access to the pre-trained embeddings (Wolf et al., 2020). We use PyTorch Lightning to organize the training code (Falcon, 2019). We fix all hyperparameters, which are reported in Table 6.

We want to call attention to BERT requiring much less data than T5 to capture our target features. At face value, it seems that BERT requires much less data than T5 to capture our target features. However, we are wary about making such strong claims. Something to consider here (noted in the Appendix B is that for T5 we used a linear model rather than formatting the task in text (which is how T5 is trained). We made this decision (1) because we had trouble training T5 in this purely textual manner, and (2) using a linear classification head over two classes is consistent with the other model architectures. Again, GPT2 and RoBERTa performed on par with BERT, so the difference between the performance of BERT and T5 may be due to how we trained T5.

## C    MEASURING EXTRACTABILITY INDIRECTLY

We measure the MDL for $t$ with *both* and $s$-only examples. In the simulated setting we can compare this approach with measuring the MDL directly ($t$-only vs *neither*). See Table 7 for MDL results. The ordering of the feature's difficulty holds across the two methods.

## D    TEMPLATES FOR NATURALISTIC DATA

Each template corresponds to a combination of target features, grammars, and spurious features (the target and spurious features are discussed in Section 4.1). See Table 8 for a complete list of templates. See Table 10 for further details about the templates that are used for each of the the target features. Complete details about implementation of these templates (and all data) will be released upon acceptance.

## E    WHY EXACTLY IS IT HARD TO GENERATE T-ONLY EXAMPLES?

Target features may be unavoidably linked to spurious ones. For example, for a Negative Polarity Item to be licensed (perhaps smoothing over some intricacies) the NPI ("any", "all", etc) must be a downward entailing context. These downward entailing contexts are created by triggers, e.g., if a negative word like "no" or "not" or a quantifier like "some". Linguists who study the problem have assembled a list of such triggers (see Hoeksema (2008)). Arguably, one cannot write down a correct example of NLP licensing that doesn't contain one of these memorizable triggers. Thus, we cannot train or test models on correct examples of NPI usage while simultaneously preventing it from having access to trigger-specific features.

| Hyperparameter | Value |
|---|---|
| *Hyperparameters* | |
| random seed | 1, 2, 3 |
| batch size | 128 |
| cumulative mdl block sizes (%) | 0.1, 0.1, 0.2, 0.4, 0.8, 1.6, 3.05, 6.25, 12.5, 25 |
| $s$-only rates (%) | 0, 0.1, 1, 5, 10, 20, 50 |
| *T5 Hyperparameters* | |
| model keycode | t5-base |
| model architecture | T5Model |
| warmup | linear (default values) |
| optimizer | AdamW |
| pooler | last hidden state |
| lr | 2e-5 |
| batch size | 128 |
| classifier head | Linear |
| *BERT/RoBERTa/GPT2 Hyperparameters* | |
| model keycode | bert-base-uncased |
| model architecure | BertForSequenceClassification |
| warmup | cosine (default values) |
| optimizer | AdamW |
| pooler | first hidden state |
| lr | 2e-5 |
| batch size | 128 |
| classifier head | MLP |
| *LSTM (Toy and GloVe) Hyperparameters* | |
| optimizer | Adam |
| pooler | last hidden state |
| lr | 1e-3 |
| batch size | 128 |
| classifier head | MLP with 1 hidden-layer tanh activation |
| hidden size | 300 |

Table 6: **Hyper and System Parameters.** We use Hugging Face for the underlying model implementations.

| Feature | Probe | MDL in $k$-bits |
|---|---|---|
| adj-dupl | ($s$-only vs *both*) | 242.19 |
| | ($t$-only vs *neither*) | 248.68 |
| contains-1 | ($s$-only vs *both*) | 0.29 |
| | ($t$-only vs *neither*) | 0.34 |
| first-last | ($s$-only vs *both*) | 397.64 |
| | ($t$-only vs *neither*) | 607.46 |
| prefix-dupl | ($s$-only vs *both*) | 175.74 |
| | ($t$-only vs *neither*) | 193.92 |

Table 7: We measure the target MDL directly on the toy data, where we can access the target feature. Recall that in natural conditions we can not generate examples with the target feature (free of spurious features), so we used a dataset comprising *both* and $s$-only examples. In the simulated setting, our approach for measuring the target extractability indirectly by using *both* and $s$-only examples reports results similar to those when directly using *target* and *neither*. These values are in $k$bits.

| Target feature | Spurious features | grammar | # |
|---|---|---|---|
| SVA | closest-noun, length[1], lexical, plurality | base, nested | 7 |
| NPI | length, lexical, plurality, tense | base | 4 |
| GAP | length, lexical, plurality, tense | base, islands[2] | 9 |

[1]Length is not applicable for the base template

[2]We also run the islands template without additional spurious features.

Table 8: List of templates that are used in Section 4. These are discussed in greater detail below.

| Target | Spurious | Example |
|---|---|---|
| Subject agrees with verb | N before V is singular | [*neither*] * The piano teachers of the lawyers wounds the handyman. |
| NPI in down.-entailing context | Contains negation word | [*neither*] * The student who was wrong ever resigned. |
| Correct filler-gap dependency | Main verb is in past tense | [*neither*] * I know that he recognized __ yesterday. |

Table 9: *neither* examples for Table 2.

| Subcase | Template | Example |
|---|---|---|
| NPI ✓ | No $N_{p-neg}$ ever $V$. | No girl who was sad ever resigned. |
| NPI ✓ | $Det\ N_p\ V$. | Some student who no girl ever followed ran. |
| NPI $X$ | The $N_p$ ever $V$. | *The man who some lawyer hated ever traveled. |
| NPI $X$ | The $N_{p-ever}\ V$ | *The boy who ever smiled shouted. |
| SVA ✓ | The $N_1^x$ of the $(N_i$ of the $)^*\ N_2\ V_1^x$ the $N_3$. | The piano teacher of the lawyers wounds the handyman. |
| SVA $X$ | The $N_1^x$ of the $(N_i$ of the $)^*\ N_2\ V_1^y$ the $N_3$. | *The piano teacher of the lawyers wound the handyman. |
| | *Whether or not a noun and verb agree is given by whether their superscripts match.* | |
| GAP ✓ | $(N_i\ V_i$ that$)^*\ N_1\ V_1$ who $(N_i\ V_i$ that$)^*\ (N_i\ V_i)$. | I know who he believed. |
| GAP ✓ | $(N_i\ V_i$ that$)^{1+}\ N_2$. | I know that he believed them. |
| GAP $X$ | $(N_i\ V_i$ that$)^*\ N_1\ V_1$ who $(N_i\ V_i$ that$)^*\ N_1\ V_1\ N_2$. | *I know who he believed them. |
| GAP $X$ | $(N_i\ V_i$ that$)^*\ N_1\ V_1$. | *I know that he believed. |
| ISL $X$ | $N_1\ V_1$ who $N_2\ V_2\ N_4$ that $N_3\ V_3$. | *I know who he knew the lawyers that she believed. |

Table 10: List of templates that are used in Section 4. These do not include the spurious features which are discussed in Section 4.1. For NPI, each $N_*$ represents a noun phrase. $N_{p-neg}$ is valid after a negation (might contain a polarity item). $N_p$ is valid after a determiner (cannot contain an unlicensed polarity item). $N_{p-ever}$ is not valid after a determiner (contains an unlicensed polarity item). These phrases have complex nesting behavior and can become arbitrary long. In addition, a sentence might consist of multiple independent clauses, each of which is given by one of these templates. For SVA, the base templates do not have the additional nouns in the starred parentheticals, while the nested templates have zero or more. For GAP, the harder set of templates include ISL (island) examples as an additional $s$-only example that force the model to not violate (one specific type) of island constraint. For complete details and lexicons see the source.

Similar to the NPI example, it's not possible (to our knowledge) to construct target-only examples for filler-gap since construction requires a wh-word and syntactic gap; thus, we can't create a positively labeled, grammatical sentence that exhibits a Filler Gap without these elements.

In summary, target-only examples may add new spurious features (as with NPI), or be impossible to construct because the presence of the target feature implies the presence of the spurious feature (as with filler gaps). Still, our setup permits the MDL to be computed directly with target-only examples, and so, in cases where it is feasible to create target-only examples (e.g. the Subject-Verb Agreement templates), it would have bolstered our argument to do so.

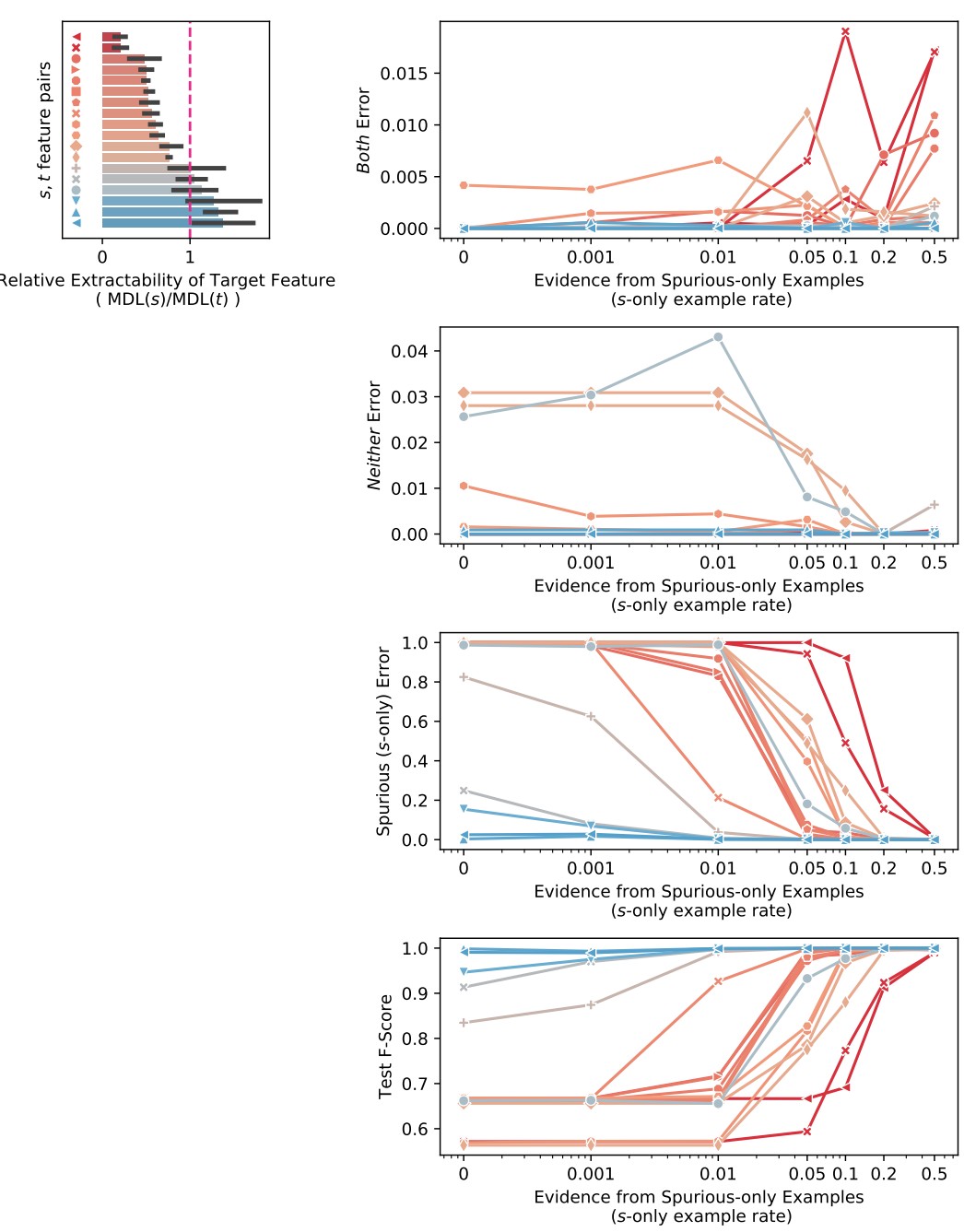

Figure 6: T5.

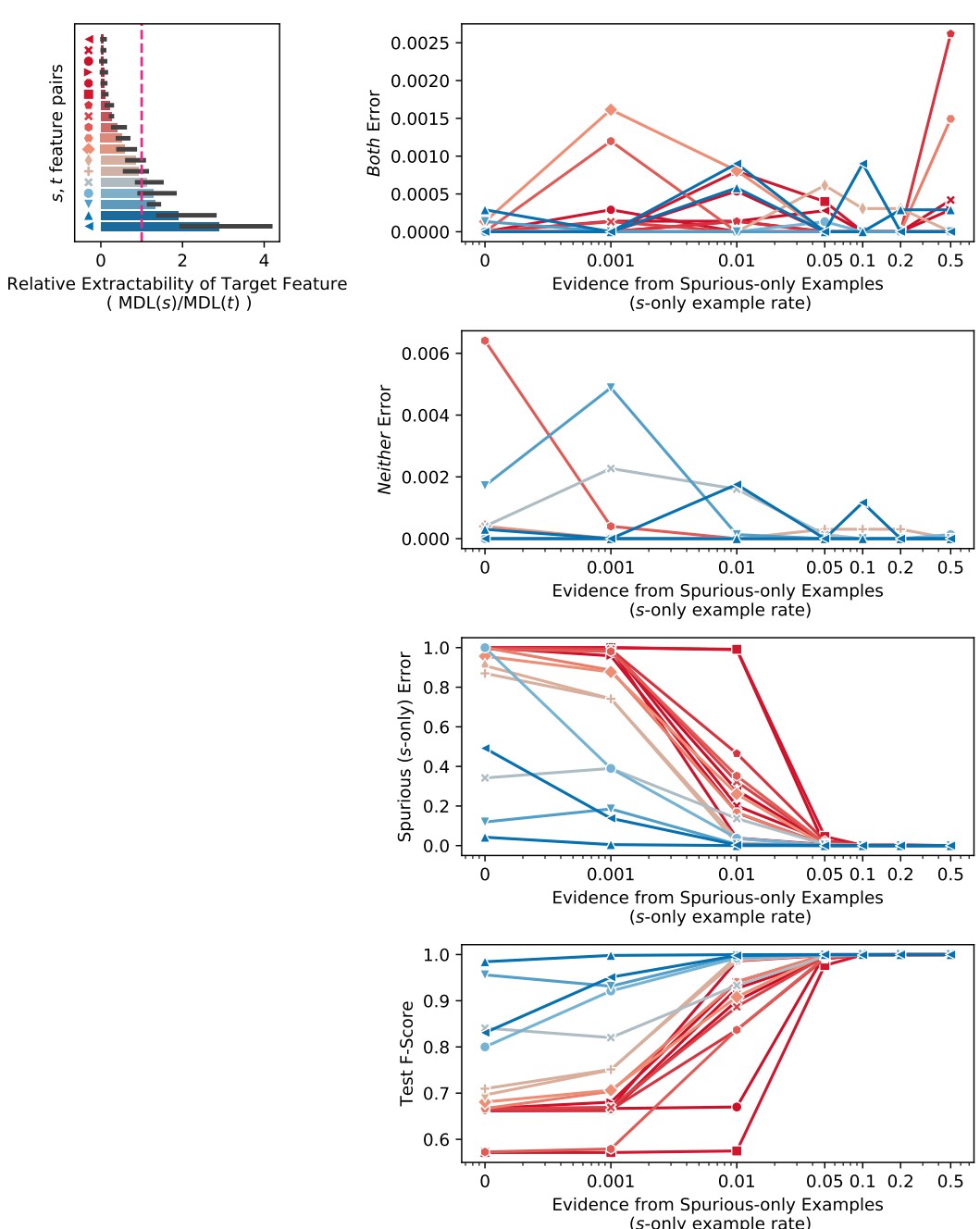

Figure 7: BERT.

# F  MDL ISSUES: OVERFITTING IN THE SYNTHETIC EXPERIMENTS

We found that the MDL exceeds the uniform code length in some of the synthetic experiments. We found that this occurs because the model overfits on the small early-block sizes. See Figure 11.

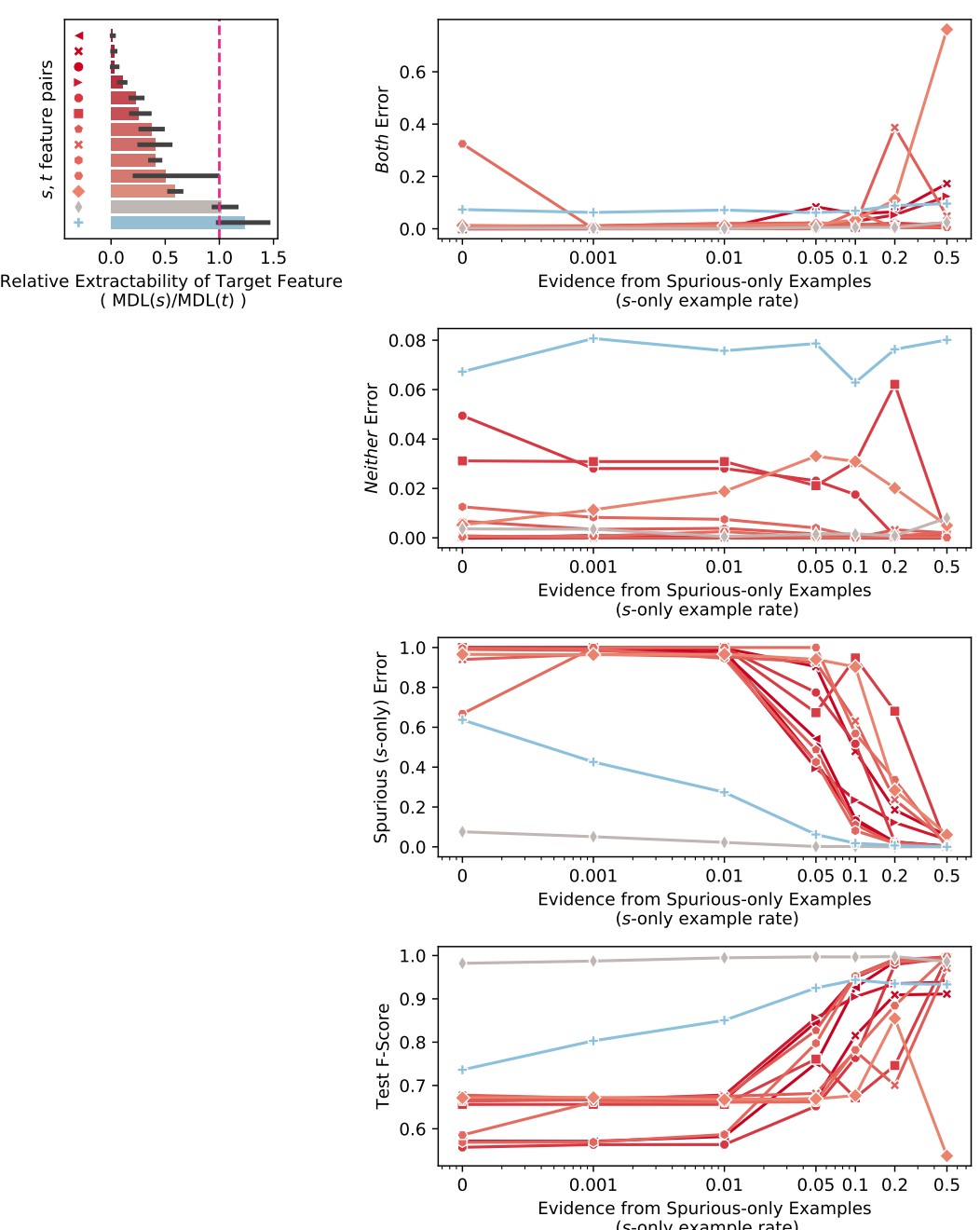

Figure 8: GloVe.

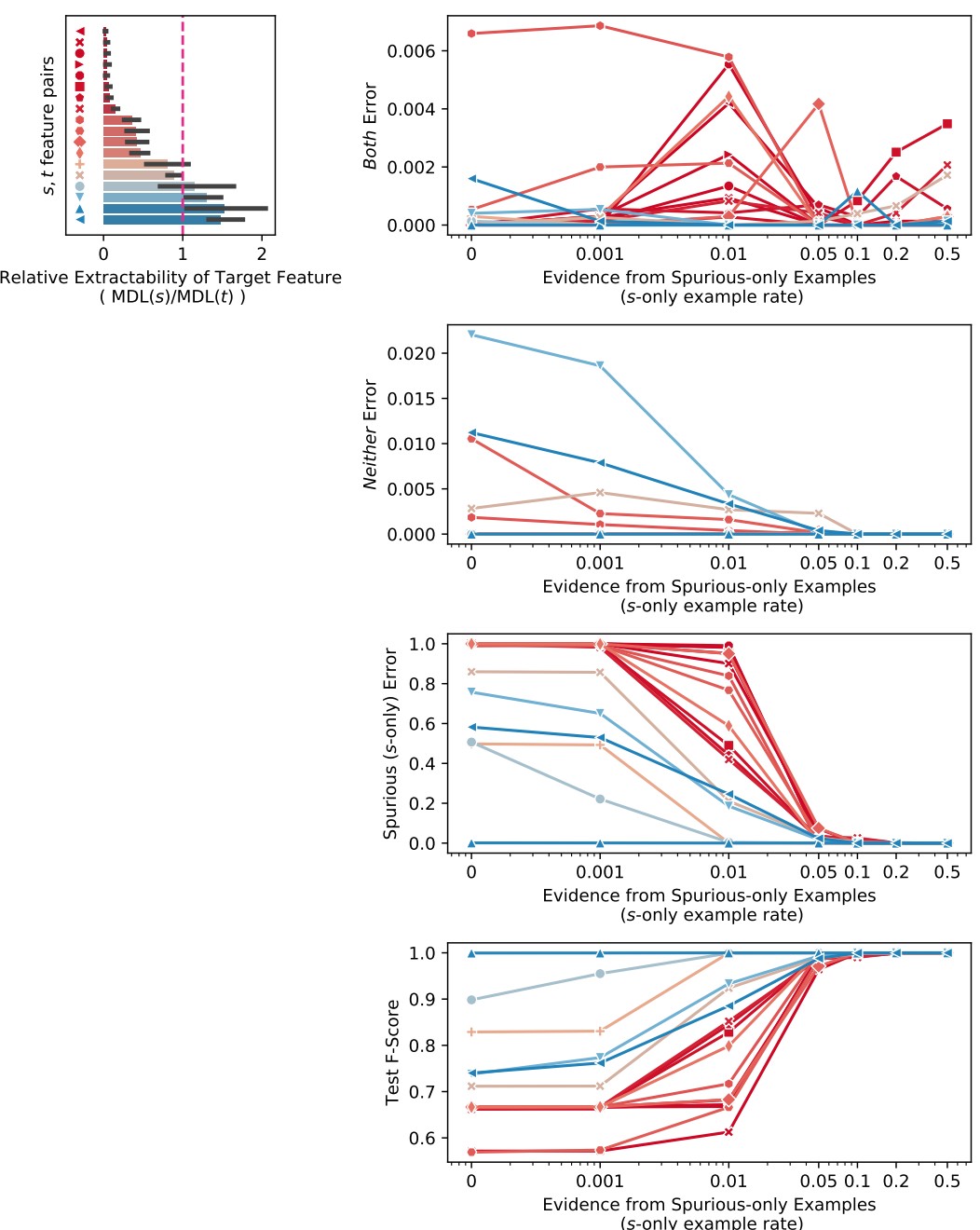

Figure 9: GPT2.

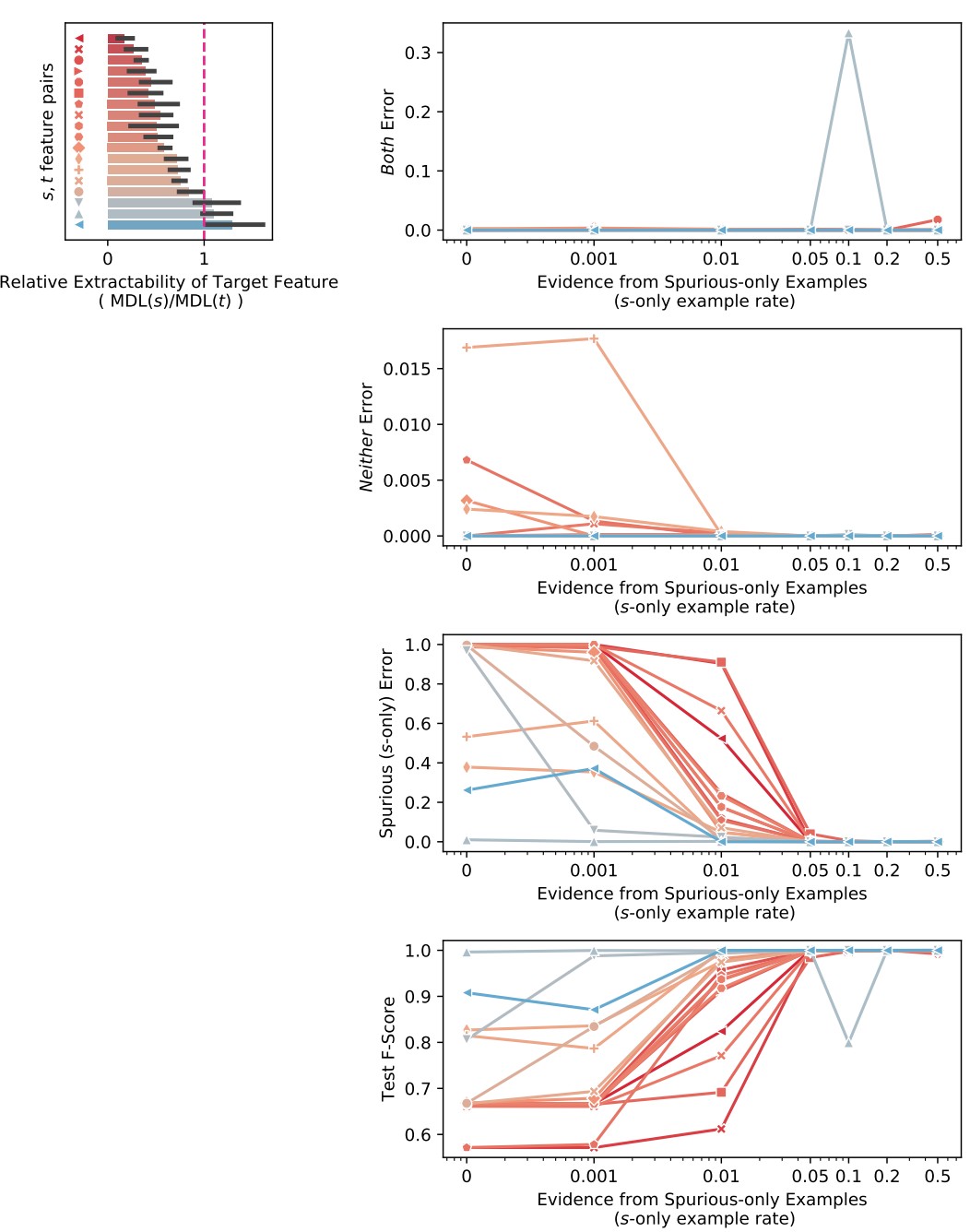

Figure 10: RoBERTa.

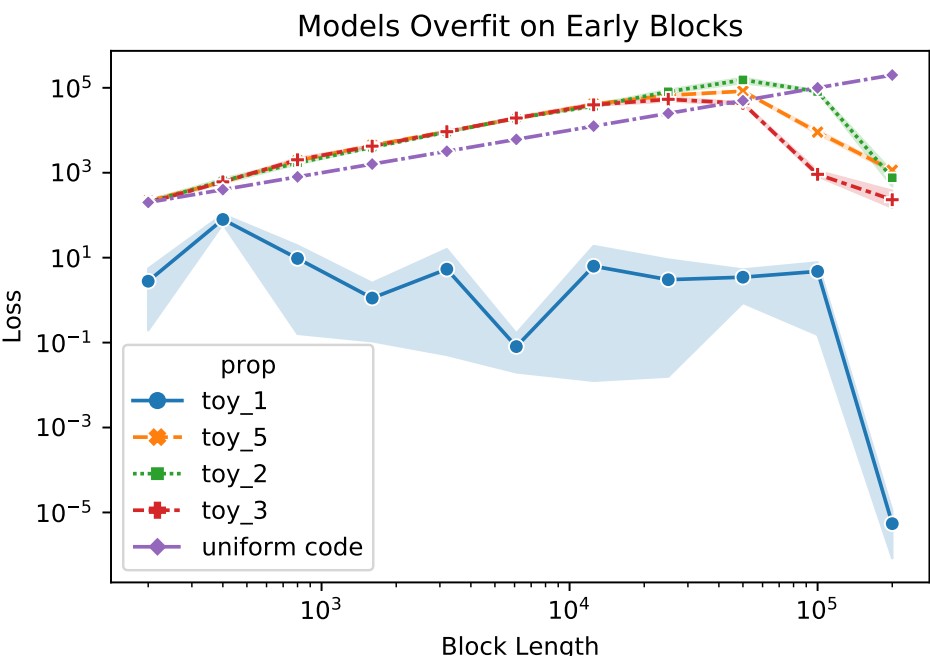

Figure 11: Overfitting on the Synthetic Tasks.

