# OpenReview forum: "Predicting Inductive Biases of Pre-Trained Models"
_ICLR.cc/2021/Conference — ICLR 2021 Poster_

### Official Review · AnonReviewer2 · 2020-10-21
**Important work filling a gap in current NLP interpretability literature**

**Rating:** 8
**Confidence:** 4

**Review:**


# Summary
This paper studies the relationship between extractability of features from pre-trained representations and how much a fine-tuned model uses that feature. The extractability of features is measured by the minimum description length of a probing classifier trained to detect the feature from the pre-trained representations (using the online code version of Voita and Titov). The degree to which a fine-tuned model uses the feature is measured by the amount of evidence required for a model to tease apart spurious from non-spurious features (called "target" features). Evidence here means examples where a spurious feature occurs but a non-spurious feature does not occur. When there are many such examples (high spurious-only rate), it is easier for a model to reject the spurious feature and learn to rely on the target feature. The "degree to which a fine-tuned model uses a feature" is defined as the minimal spurious-only rate at which the model can accomplish the task.

The paper has two kinds of experiments, on synthetic and more natural data. The synthetic data are sequences of symbols where the task is to identify simple properties like occurrence or repetition of symbols. The experiments are set up such that varying rates of spurious-only examples are presented during training, providing increasing amounts of evidence against the spurious feature (presence of the symbol 2) and in favor of the target feature. The target feature is identical to the label, that is, it is 1 when the example corresponds to the label and 0 otherwise. The paper reports extractability of the spurious and target features via the MDL of a probing classifier. The metric of interest is the relative MDL, where higher means the feature is more extractable. When the features are more extractable, less evidence is required for the model to reject spurious features. With less extractable features, more evidence is required.

The natural language examples are made with acceptability judgements of examples generated by grammars for three linguistic phenomena (subject-verb agreement, negative polarity items, and filler gap dependencies). Here again the setting is similar, modulu a tweak on how to calculate extractability. The main result here is high (negative) correlation between extractability and evidence required for rejecting the spurious feature.

# Main comments
1. This paper fills an important gap in the NLP interpretability literature that has recently been a cause of concern in the community. On the one hand, probing classifiers tell us something about the existence (and more recently the extractability) of properties in pre-trained models' representations. But they do not tell us whether a model uses those properties. One the other hand, many challenge sets and test suites tell us whether a model can successfully perform a task requiring some linguistic property. The paper aims to connect these two aspects, and it does so quite convincingly, although I have some reservations below.
2. The experimental setup is well designed. The use of synthetic data allows a fairly clean setup where spurious and non-spurious features are distinct and simple. The experiments of training with increasing amount of spurious-only examples are instructive.
3. The natural language examples are important as they go beyond synthetic data and closer to a naturalistic scenario. However, these are still templatic sentences and synthetic in a sense. I wouldn't call these naturalistic examples. Ideally, experiments on naturally occurring data would be more convincing. Or, at the very least a discussion of this issue should be made.
4. The paper makes use of recent advances in interpretability work, including information-theoretic probing, and draws connections to a broad range of related work.
5. The assumption that the extent to which a model uses a feature can be measured by the spurious-only error rate (at some spurious-only occurrence rate) is questionable in my opinion. In a very clean setting like the synthetic data, I could maybe accept it. But, "using" is in fact a causal concept, while a causal mechanism has not been demonstrated. The paper alludes to this point in the discussion, but I think the discussion around this point should be expanded, and the strong claims should be rephrased or modulated.


# Questions and other comments
1. The paper makes the assumption that the target feature t and the label are the same. I am not convinced about the "without loss of generality" claim. In practice, it is not easy to isolate a feature t that is identical with the label. How would this assumption affect the generalization of the approach to more realistic scenarios?
2. The task is a binary classification task. The features holding is also binary, that is either a feature holds (1) or not (0). But, suppose the label is 0, then the t feature is also 0, meaning it does not hold. This seems contrary to what is meant. This could be a confusion on my part.

## Synthetic data
3. Why is MDL computed by training a classifier to distinguish s-only from neither, and not from some other part of S?
4. Footnote 3 is concerning - Aren't MDLs higher than a uniform code meaningless?
5. The classifier is not so simple (LSTM + 1-layer MLP). Why is that? How does the identity of the classifier affects the results?
6. The both-error subplot in figure 2 shows a slight increase in error rate with large s-only rate. Does this mean that the model has (falsely) learned to reject the example when s is in it? That is, it has learned another spurious feature, just in the other direction, instead of learning to rely on t.
7. A similar pattern is found in the t-only error subplot. There, even with high s-only rate, the models don't classify t-only examples correctly. I wonder why this plot is different from the s-only error plot, as this shows a directional behavior. Some discussion of this would be useful.
8. There seems to be a stark contrast between the contains-1 feature and the other three, both in terms of MDL and in figure 2. Is it possible to show a more gradual behavior between the two extremes?

## Natural language examples
9. Why are the training sets so small? How does this affect MDL numbers and their validity? Apropos footnote 1.
10. Why exactly is it hard to generate t-only examples? The appendix is indeed helpful in making sure the MDL(t) calculation method is legit, but more clarification around this issue would be good.
11. Here, s-only error is used as "the use of the spurious feature", but this is only one aspect in which a model may make use of s. It may be that a model makes more complicated use of s, when s is found in combination with t. The discussion touches upon this point by acknowledging that the work does not establish a causal relationship between extractability and feature use. I'd go even further and say that "feature use" should be defined in causal terms.
12. What is the performance (F-score) for determining s-rate*? is that the performance on s-only examples? on other examples? Why the shift to F-score now?
13. Discuss y-axis differences in figure 3c. BERT needs much less evidence than (some cases of) GloVe and T5. How does that impact the analysis?
14. The term "learning curves" for figure 4 is confusing: those aren't results during training, right? They are results after training, each time with a different rate of s-only examples.

---

### Official Review · AnonReviewer3 · 2020-10-27
**The paper aims to bridge the gap between model interpretation using probing and model's use of spurious features. They show that the findings of MDL with respect to a feature correlate with the extractability of the feature, given the evidence of representing the feature is available in the training data. The results are presented using both synthetic and natural language data.**

**Rating:** 7
**Confidence:** 2

**Review:**

The paper aims to bridge the gap between model interpretation using probing and model's use of spurious features. They show that the findings of MDL with respect to a feature correlate with the extractability of the feature, given the evidence of representing the feature is available in the training data. The results are presented using both synthetic and natural language data.

I really like the premise of the paper, which is connecting the research on the linguistic learning of a model with the presence of important and spurious features in the data.

One issue I have with the work is the simplistic assumptions that are likely to be different in the real-world data. Real-world data may have various spurious features and it is possible that not one feature alone is playing a role in pushing the model to rely on spurious features. It can be a combination of spurious features plus the relative presence of important features. It is hard to imagine how this method will scale to real-world datasets. I would like the authors to comment on it.

Moreover, the findings are quite expected. In general, the probing methods including MDL were mainly aimed at analyzing the linguistic learning of the representations. In that case, MDL is scoring the representations with respect to various linguistic properties. Here, the authors are using MDL to look at how input features are represented in the model. Statistically, MDL is likely to look at the same things as the model is looking at since both of them are based on the same training data and input features. Please comment on this, in case I misunderstood the point.


Minor comments:
- what is the reason for low performance when using Glove?

---

### Official Review · AnonReviewer1 · 2020-10-28
**Clearly defined hypothesis, but limited contribution; finding seems to support existing knowledge/assumptions**

**Rating:** 6
**Confidence:** 4

**Review:**

After reading author responses:

Thank you to the authors for your detailed responses. With regard to the highlighted implication that "the harder feature can be obscured completely by a spurious one; i.e., there are settings in which the model just won't adopt the harder feature at all" --  to clarify, while my phrasing may not have made this apparent, I was assuming this implication in my interpretation of the results. So my impression of the finding is not changed substantially by the author response. However, I do want to give appropriate acknowledgment of the value of explicitly testing/confirming intuitive explanations of model behaviors, and it is clear that other reviewers find value in the contribution, so I am bumping my score up a bit.

~~~~~~~~~~~~~~~~~~~~~~~~~~~~~~~

This paper addresses a seeming contradiction between findings that indicate encoding of linguistic information in models' internal representations, and findings that show models not to use more sophisticated linguistic information during fine-tuning on downstream tasks. The paper hypothesizes that a model's use of a given feature can be explained as a function of extractability of the feature in combination with the amount of evidence for that feature's predictive reliability. The authors test on toy, non-language data as well as natural language data, and find support for their hypothesis.

All in all I think this is a reasonably clear and well-written paper, with a concrete and intuitive hypothesis. My main concern is that the motivating issue is a bit of a strawman, in that the posited explanation was fairly obvious as a means of reconciling the "contradiction" raised at the start of the paper. I can't speak for the rest of the community, and it may be that this is something that people have found puzzling -- but speaking for myself I can say that I haven't at any point considered the highlighted "contradiction" to be a contradiction, having simply assumed something like the explanation hypothesized in this paper. Now, there is of course value in providing concrete evidence supporting intuitive assumptions made by the community. However, as the authors point out,  related intuitions have already been supported by, e.g., evidence that models will more readily pick up on "easy" examples over "difficult" examples. So it's not clear to me that the paper is making a sufficiently novel, surprising contribution at present.

I think one way in which these findings would be more compelling would be if the measure of extractability were defined independently of empirical classifier sensitivity. As it is, the experiments are seemingly demonstrating that the more readily a classifier is able to pick up on a given feature, the more readily another classifier will use that feature during learning.  I have to assume that this will strike most readers as obvious. However, if extractability/MDL were measured independently of classification performance, then we would presumably learn some interesting and valuable things about what determines extractability for these models.

Smaller notes:

The two sets of experiments are described as "synthetic" versus "natural language" -- but if I'm understanding correctly, the natural language examples are generated synthetically. If this is correct, then the current framing of the distinction is misleading.

Figure 2 is difficult to interpret, and the placement of the legend is odd-looking and confusing. Fig 3 is also pretty difficult to extract information from -- generally presentation of information could be made clearer for the reader.

The wording on p3 can be taken to imply that MDL was introduced by Voita & Titov (2020). I would recommend rephrasing and/or also citing earlier MDL references.

---

### Author Response · Authors · 2020-11-17
**Our findings are non-obvious and have important implications**

Thank you for your detailed reviews. In this (and the next) response we focus on the high-level concerns of R1/R3. We’ll address the itemized concerns of R2 in a separate response.

@R1/@R3 ask if our hypothesis is too obvious a finding. We believe our hypothesis is intuitive, but it is not tautologically true.

@R1: “The experiments are seemingly demonstrating that the more readily a classifier is able to pick up on a given feature, the more readily another classifier will use that feature during learning.”
@R3: “...the findings are quite expected… MDL is likely to look at the same things as the model is looking at since both of them are based on the same training data and input features.”

We find evidence for the hypothesis: The more extractable a target feature is compared to a spurious feature, the less training evidence needed for a model to use the target feature (and generalize). An implication of this is that the harder feature can be obscured completely by a spurious one; i.e., there are settings in which the model just won't adopt the harder feature at all. This is different from the alternative interpretation to which the reviewers refer, in which the harder feature is learned later than the spurious one, but is still learned.

Indeed, we find that BERT & T5 are both entirely capable of learning the target feature: both perfectly solve the classification task when the label can be predicted only from the target feature or only from the spurious feature (See footnote 1). But, even though the harder (target) features are present (and learnable), the models do not learn them when the target feature is relatively more difficult to extract than the spurious features.

Thus, @R1’s paraphrase that “the more readily a classifier is able to pick up on a given feature, the more readily another classifier will use it” misses a crucial insight. Our results show that if one classifier does not pick up on a feature readily enough, another classifier (or, rather, the same classifier trained with different data) may not be sensitive to  that feature at all. (And, equally importantly, if a feature is readily-enough detected, the classifier will be sensitive to it even without overt statistical incentive to prefer it. This is a type of inductive bias that is very desirable for many language learning problems.)  This finding changes how we view fine-tuning, which is generally considered to be beneficial because it allows models to learn new, task-relevant features. Our findings suggest that if the needed feature is not already “readily enough available” after pre-training, fine-tuning will not have the desired effect.

We will include discussion on these points in the paper, highlighting the distinction between this work and previous findings.

(Footnote 1):  *BERT, T5, get >0.99 accuracy on all feature pairs when testing the target feature in isolation. This control study also illustrates why the glove-based LSTM failed: The GloVe model only solved the target task in isolation for only 60% (12/20) feature pairs.*

---

### Author Response · Authors · 2020-11-17
**We contribute to an active research direction in NLP**

This response focuses on the concerns of @R1/@R3.

There is active interest in the problems we explore in this submission. Concurrent work [1], which we became aware of after submission, investigates when the inductive biases of large pre-trained models (RoBERTa) shift from a surface-based feature (what we call spurious features) to a linguistic-based feature (what we call a target feature). In our work, we further show how to predict which of these two biases most characterize the model and the generalizations it acquires . Our approach is not analytic, but still, we are able to predict this inductive bias without evaluating on a downstream task . From a technical perspective, this work connects the recent wave of probing results with actual model behavior, which has previously been wanting. E.g., another recent work, [2] strives to close this loop by connecting structural analysis (of which probing is an example) to behavioral analysis (model predictions), by finding neurons that are causal with respect to the model predictions.

@R3 “Real-world data may have various spurious features and it is possible that not one feature alone is playing a role in pushing the model to rely on spurious features.”

We agree that real-world data will not always have simple target and spurious features. However, our definition of a feature accommodates multiple spurious or target features. We could construct the spurious or target feature in our setup to be a combination of several features. In fact, some of our spurious features are already defined in this way: the lexical feature, for example, is defined as a combination  of several individual-word features because it holds if one of a set of words is in the sentence. This type of spurious feature is common in real datasets: E.g., the hypothesis-only baseline in NLI is a disjunction of lexical features (with semantically unrelated words like “no”, “sleep”, etc.) [3, 4].

There are important differences between our setup and naturally occurring data, but we believe it’s important to establish a relationship between extractability and downstream feature usage in a relatively simple setting as a precursor to exploring more complex hypotheses. Still, we appreciate your point and will include discussion of the assumptions of our setting in the final paper.

[1] Warstadt, Alex, et al. "Learning Which Features Matter: RoBERTa Acquires a Preference for Linguistic Generalizations (Eventually)." arXiv preprint arXiv:2010.05358 (2020).
[2] Vig, Jesse, et al. "Causal mediation analysis for interpreting neural nlp: The case of gender bias." arXiv preprint arXiv:2004.12265 (2020).
[3] Gururangan, Suchin, et al. "Annotation artifacts in natural language inference data." arXiv preprint arXiv:1803.02324 (2018).
[4] Poliak, Adam, et al. "Hypothesis only baselines in natural language inference." arXiv preprint arXiv:1805.01042 (2018).

---

> ### Comment · AnonReviewer2 · 2020-11-17
> **Refer to reviewers by their designated IDs**
>
> Hi, it seems like you're numbering reviewers by the order in which the reviews appear, while it would be better to use the reviewer number assigned by OpenReview. So, the order is 1,2,3, but the assigned IDs are actually 1,3,2. This would help navigate the comments.

---

> > ### Author Response · Authors · 2020-11-17
> > **Fixed IDs**
> >
> > Fixed — thanks!

---

### Author Response · Authors · 2020-11-24
**Additional Points**

## (1/7)

In this response (and the following) we respond to the additional points raised by the reviewers. We also itemize the corresponding changes in our paper.
### Paper Changelist
- Incorporated some of the high-level responses into the paper’s discussion section.
- Added additional discussion on the assumptions of our setting in the final paper (re. real-world data).
- Updated paper with correct MDLs. In the second set of experiments (Section 4), they were scaled down by approximately the batch-size.  This changed the correlations slightly, but did not change any patterns or conclusions.
- Updated Figure 2 to move the legend to the bottom.
- Updated Figure 3 to give each subplot its own row.
- Updated Figure 3: we now use the same scale for each subplot, share the axis labels, and use log scales.
- Updated footnote regarding the definition of a target feature.
- Supplemented the results with experiments on GPT2 and RoBERTa; added discussion of differences between T5 and other models.
- Updated the description of the “learning curve”.
- Added a figure showing the overfitting on early blocks for MDL on the first set of experiments (Section 3) to the Appendix.
- Fixed a typo that indicated we had 16 not 20 feature pairs in the Appendix.

---

> ### Author Response · Authors · 2020-11-24
> **(7/7)**
>
> ### Reviewer #2 (continued)
>
> ***Discuss y-axis differences in Figure 3c. BERT needs much less evidence than (some cases of) GloVe and T5.***
> At face value, it seems that BERT requires much less data than T5 to capture our target features. However, we are wary about making such strong claims. Something to consider here (noted in the Appendix hyperparameters) is that for T5 we used a linear model on top of T5’s pretrained encodings, rather than formatting the task in text (which is how T5 is trained). We made this decision (1) because we had trouble training T5 in this purely textual manner, and (2) using a linear classification head over two classes is consistent with the other model architectures.
> We added further experiments with GPT2 and RoBERTa. They performed on par with BERT, so the difference between the performance of BERT and T5 may be due to how we trained T5. We added the GPT2 and RoBERTaresults to the main body of the paper, and added a brief discussion in the paper about this possible explanation for the performance of T5.
>
> ***The term "learning curves" for figure 4 is confusing: those aren't results during training, right?***
> Correct and agreed -- we updated the terminology there.

---

> ### Author Response · Authors · 2020-11-24
> **(6/7)**
>
> ### Reviewer #2 (continued)
>
> ***The classifier is not so simple (LSTM + 1-layer MLP). Why is that? How does the identity of the classifier affect the results?***
> The features we use in the first set of experiments (Section 3) requires that our classifiers represent token order. A bag-of-words or n-gram model would, for example, not be able to learn the “first-last” pattern (whether the first token is equal to the last) and the “prefix-duplicate” pattern (whether the first two tokens are equal). The “first-last” pattern requires the model to track both the first and last token (and whether or not they match). A typical n-gram would not be able to access this information. Likewise, a count-based representation like a bag-of-words would not track what the first and last values.
> An n-gram model could have a high recall for the “prefix duplicate” patten, but would not be able to fully learn the feature: An n-gram could only detect if that there is a duplicate, but it could not disambiguate duplicates that occur in positions in the prefix verus anywhere else in the sentence.
> That all being said, we appreciate your point and acknowledge that it would be ideal if we tested more classifiers in the first set of experiments. We have added more models to our second set of experiments (GPT2 and RoBERTa) to further confirm that our results are not model-dependent. (For this second set of experiments we ran pilot studies with BOW and CNN-based models, and found that both were unable to solve the task. We make note of this in footnote 5.)
>
> ***Here, s-only error is used as "the use of the spurious feature", but this is only one aspect in which a model may make use of s… The discussion touches upon this point by acknowledging that the work does not establish a causal relationship between extractability and feature use. I'd go even further and say that "feature use" should be defined in causal terms.***
> We agree that feature-use ought to be defined in causal terms, and we’re interested in making this connection. That being said, in this paper we did not do this, nor did we mean to suggest that our work was establishing a causal relationship. A better term might be “compatible” or “consistent”. As mentioned above, we plan replace the phrase  “feature usage” with something more carefully framed, and will add this discussion to the camera ready.
> We also agree that s-only error only captures “one aspect in which a model may make use of s.” Describing the s-only error as “the use of the spurious feature” was intended to develop the reader’s intuition of our findings and goals. However, for the camera-ready version, we will add discussion that s-only error does not in itself represent “use of the spurious feature” and we will point to the results on the other partitions of the data in the Appendix.
>
> ***What is the performance (F-score) for determining s-rate★? Is that on s-only examples, other examples? Why the shift to F-score now?***
> We set the threshold to be a 0.99 F-score across the test set, which includes spurious-only, neither, and both examples. This is written in the body of the text (Section 4.3), and we now include it in Figure 3 as well. We shift to F-score (rather than accuracy or spurious-only error) to ensure that the model is performing well on all available sections of the data.
> Figure 2 asks if the relative MDL correlates with s-rate★. At an F-score of 0.99 the model is consistent with the target feature (on the available partitions of the dataset).  Using instead the spurious-only error, for s-rate★, could be misleading, as the spurious-only error could be low, but all other partitions of the data could have high error. We were doubly concerned about this because s-rate★ is coarse, aggregating information across multiple runs. This makes it harder to spotcheck or visualize. Lastly, we use F-score not accuracy because the labels are not balanced on the test set (as we have 1000 each of both, neither, and spurious-only).
> On the other hand, we show the s-only error in the line plots (Figure 4) (question above) because it’s easier to interpret than an aggregated metric. Furthermore, the s-only error is a fair representation of the model performance because the models generally perform well on the other partitions (we show all results in the Appendix).

---

> ### Author Response · Authors · 2020-11-24
> **(5/7)**
>
> ### Reviewer #2 (continued)
>
> ***Why are the training sets so small? How does this affect MDL numbers and their validity? Apropos footnote 1.***
> Our templates (grammars) and lexicons do not allow us to generate datasets much larger than what we use. A larger dataset would be better, although small datasets are not uncommon in the pre-train/finetune paradigm.
>
> MDL can be thought of as the cost of encoding a dataset, so a smaller dataset results in a lower MDL. However, we use the same dataset size for each of our experiments, accounting for this effect.
>
> Additionally, in most cases, the models have good classification performance with access to the complete training set, so we hypothesize that the cost of encoding additional, larger blocks will be relatively low and will not significantly affect the MDL. We find evidence for the limited effect of larger blocks in experiments that we have added to the Appendix where we investigate the cost of encoding each block (for the features in Section 3, Figure 1,2).
>
> ***Why exactly is it hard to generate t-only examples? The appendix is indeed helpful in making sure the MDL(t) calculation method is legit, but more clarification around this issue would be good.***
> Target features may be unavoidably linked to spurious ones. For example, for a Negative Polarity Item to be licensed (perhaps smoothing over some intricacies) the NPI (“any”, “all”, etc) must be a downward entailing context. These downward entailing contexts are created by triggers, e.g., if a negative word like “no” or “not” or a quantifier like “some”. Linguists who study the problem have assembled a list of such triggers (see [Hoeksema (2012)](http://www.let.rug.nl/hoeksema/NPI-types.pdf]). Arguably, one cannot write down a correct example of NLP licensing that doesn’t contain one of these memorizable triggers. Thus, we cannot train or test models on correct examples of NPI usage while simultaneously preventing it from having access to trigger-specific features.
>
> Similar to the NPI example, it's not possible (to our knowledge) to construct target-only examples for filler-gap since construction requires a wh-word and syntactic gap; thus, we can’t create a positively labeled, grammatical sentence that exhibits a Filler Gap without these elements. Similar arguments hold for learning in general, not just NLP. E.g., [Carey (2009)](https://link.springer.com/article/10.1007/s11191-010-9307-2)  makes the argument about how monkey’s learn eyetracking: “...every time eyes are pointed at an object, so are mouths and noses, yet...monkeys avoided the competitor that was looking at the grape, not whose mouth was pointing at the grape.”
>
> In summary, target-only examples may add new spurious features (as with NPI), or be impossible to construct because the presence of the target feature implies the presence of the spurious feature (as with filler gaps). Still, our setup permits the MDL to be computed directly with target-only examples, and so, in cases where it is feasible to create target-only examples (e.g. the Subject-Verb Agreement templates), it would have bolstered our argument to do so. We plan on including this in-depth discussion in the camera-ready appendix.

---

> ### Author Response · Authors · 2020-11-24
> **(4/7)**
>
> ### Reviewer #2 (continued)
>
> ***Footnote 3 is concerning - Aren't MDLs higher than a uniform code meaningless?***
> We include experiments in the (Appendix F, Fig 10) that indicate that these high MDL’s are indeed the result of overfitting by the model on the early blocks. As discussed in Footnote 3, the information-theoretic interpretation of these cases is that the model learns an encoding that is worse than the uniform baseline on these blocks. We appreciate your point here; if the MDL’s are higher than the uniform code, then they are no longer “minimum” in some sense.
>
> However, for the sake of comparison to other work, we follow the existing definition of the prequential code. Additionally, the model’s overfitting is a potentially useful signal to include in our definition of extractability, as it indicates that small amounts of data mislead the model.
>
>
> ***The both-error subplot in figure 2 shows a slight increase in error rate with large s-only rate. Does this mean that the model has (falsely) learned to reject the example when s is in it? That is, it has learned another spurious feature, just in the other direction, instead of learning to rely on t.***
>
> ***A similar pattern is found in the t-only error subplot. There, even with high s-only rate, the models don't classify t-only examples correctly. I wonder why this plot is different from the s-only error plot, as this shows a directional behavior. Some discussion of this would be useful.***
> Indeed, the model in the first set of experiments (Section 3) seems to adopt new incorrect heuristics before learning the target feature . As you point out, the model incorrectly classifies some “both” examples, evidence that it is starting to learn the incorrect heuristic that the presence of the spurious feature implies a negative label.
> We also agree the behavior in the t-only plot is interesting. The high t-only error could indicate that the additional “s-only” examples help the model become sensitive to the target feature in some cases (when the spurious feature is present) but not others (when the spurious feature is not present).
> From the perspective of the training data available to the model, both of these hypotheses are equally supported:
>
> 1. Target ↔ 1
> 2. (Target ^ Spurious) ↔ 1
>
> The second hypothesis (2) is wrong, but because we do not provide target-only examples, it is not disambiguated from (1). It is unclear when the model would (or should) choose (1) over (2). Assuming high performance on the other partitions, the target-only error indicates which of the two hypotheses the model uses (or is consistent with). Figure 2 shows that the target feature extractability orders the features’ target-only error. Our results suggest that the easier the target feature is to extract, the greater the extent a model will tend towards (1).
> We believe that both these takeaways point to the limitations of data augmentation, which is an interesting avenue for future work. While this is not the focus of this paper, we agree that it’s important to mention these patterns. We will call out the U-shaped curve in the “both” plot (though we don’t find the curves to be substantial enough to make any claims here). And we will briefly discuss the disparity between the s-only and t-only graphs that you describe and potential implications for the effectiveness of data augmentation.
>
> ***There seems to be a stark contrast between the contains-1 feature and the other three, both in terms of MDL and in figure 2. Is it possible to show a more gradual behavior between the two extremes?***
> We agree that greater variation in the first set of experiments (Section 3) would bolster our argument (we will acknowledge this in the camera ready as a direction for future work), but we are encouraged by the greater variation in the natural language experiments.

---

> ### Author Response · Authors · 2020-11-24
> **(3/7)**
>
> ### Reviewer #2
> ***Regarding the synthetic nature of the task***
> Quoting our previous comment: There are important differences between our setup and naturally occurring data, but we believe it’s important to establish a relationship between extractability and downstream feature usage in a relatively simple setting as a precursor to exploring more complex hypotheses.
>
> Still, we appreciate your points on this (regarding the templatic sentences and the nature of our target feature), and we will include more discussion on the subject in the camera-ready. For example, we will explicitly call out some of our simplifying assumptions (we have started to add this discussion in the latest version).
>
> ***The natural language examples are important as they go beyond synthetic data and closer to a naturalistic scenario. However, these are still templatic sentences and synthetic in a sense. I wouldn't call these naturalistic examples. Ideally, experiments on naturally occurring data would be more convincing. Or, at the very least a discussion of this issue should be made.***
> We appreciate the point. We will reframe our terminology for the camera-ready and add a discussion of this issue (see above note).
>
> ***The assumption that the extent to which a model uses a feature can be measured by the spurious-only error rate (at some spurious-only occurrence rate) is questionable in my opinion. In a very clean setting like the synthetic data, I could maybe accept it. But, "using" is in fact a causal concept, while a causal mechanism has not been demonstrated. The paper alludes to this point in the discussion, but I think the discussion around this point should be expanded, and the strong claims should be rephrased or modulated.***
> We agree that “use” is a causal concept, and that we’ve been using it informally. We will rephrase in camera ready to use more careful phrases such as “makes predictions consistent with feature use” , and explicitly highlight the fact that we are not making a causal argument. We discuss our use of the spurious-only rate below.
>
> ***The paper makes the assumption that the target feature t and the label are the same. I am not convinced about the "without loss of generality" claim. In practice, it is not easy to isolate a feature t that is identical with the label. How would this assumption affect the generalization of the approach to more realistic scenarios?***
>
> ***The task is a binary classification task. The features holding is also binary, that is either a feature holds (1) or not (0). But, suppose the label is 0, then the t feature is also 0, meaning it does not hold. This seems contrary to what is meant. This could be a confusion on my part.***
>
> Correct -- we make the simplifying assumption in this setting that the target feature is identical with the label. We agree that this has limitations and will include a discussion of this for camera-ready (see above note). One could extend our framework to consider more complex models of target and spurious features, but we leave this to future work.
>
> The “without loss of generality” was intended to account for features that are exactly predictive of the label but not identical (i.e. the feature which is always the opposite of the label). In this case, we can use this feature to construct a target feature that is identical to the label (by taking the opposite of the feature). We hope this clears up any confusion, and we have updated the text with some language around the w.l.o.g. claim.
>
> ***Why is MDL computed by training a classifier to distinguish s-only from neither, and not from some other part of S?***
>
> To measure extractability of the spurious feature, we chose not to train a classifier to distinguish between s-only and t-only for consistency with the second set of experiments. In that setting, generating t-only examples is difficult (see note below).
>
> Additionally, we found evidence that distinguishing between s-only and both would require sensitivity to the target feature, instead of the spurious feature (in the experiments in the Appendix that you reference).

---

> ### Author Response · Authors · 2020-11-24
> **(2/7)**
>
> ### Reviewer #1
> ***The two sets of experiments are described as “synthetic” versus “natural language” -- but if I'm understanding correctly, the natural language examples are generated synthetically. If this is correct, then the current framing of the distinction is misleading.***
> We appreciate the point and we will reframe this terminology for the camera-ready.
>
> ***Figure 2 is difficult to interpret, and the placement of the legend is odd-looking and confusing. Fig 3 is also pretty difficult to extract information from -- generally presentation of information could be made clearer for the reader.***
> We moved the legend in Figure 2 so that the charts are fully visible. We also updated Figure 3 by giving each subplot its own row and adding the axes that were previously only detailed in the captions. Beyond this, we now use the same scale for each subplot, share the axis labels, and use log scales. We are happy to make additional changes here.
>
> ***The wording on p3 can be taken to imply that MDL was introduced by Voita & Titov (2020). I would recommend rephrasing and/or also citing earlier MDL references.***
> We updated the MDL references to cite an earlier work and clarify the contribution of Voita and Titov.

---

### Decision · Program_Chairs · 2021-01-07
**Final Decision**

**Decision:**

Accept (Poster)

**Comment:**

The paper studies the features extracted by the pre-trained language model and how fine-tuning makes use of these features. The paper is well-motivated by two lines of research in the NLP area -- 1) probing approaches for understanding the features extracted in the pre-training model, 2) model behavior analysis that shows models take shortcuts for making predictions. The paper provides a comprehensive study to bridge the gaps between these two lines of discussion.

All the reviewers agree the paper has strong merits and concerns have been addressed.